# Programmable high-dimensional Hamiltonian in a photonic waveguide array

Yang Yang [1], Robert J. Chapman [1,2], Ben Haylock[3,4], Francesco Lenzini[3,5], Yogesh N. Joglekar [6] ✉, Mirko Lobino[3,7,8] & Alberto Peruzzo [1,9] ✉

Waveguide lattices offer a compact and stable platform for a range of applications, including quantum walks, condensed matter system simulation, and classical and quantum information processing. However, to date, waveguide lattice devices have been static and designed for specific applications. We present a programmable waveguide array in which the Hamiltonian terms can be individually electro-optically tuned to implement various Hamiltonian continuous-time evolutions on a single device. We used a single array with 11 waveguides in lithium niobate, controlled via 22 electrodes, to perform a range of experiments that realized the Su-Schriffer-Heeger model, the Aubrey-Andre model, and Anderson localization, which is equivalent to over 2500 static devices. Our architecture's micron-scale local electric fields overcome the cross-talk limitations of thermo-optic phase shifters in other platforms such as silicon, silicon-nitride, and silica. Electro-optic control allows for ultra-fast and more precise reconfigurability with lower power consumption, and with quantum input states, our platform can enable the study of multiple condensed matter quantum dynamics with a single device.

Waveguide arrays[1] are a powerful platform for the optical simulation of condensed matter physics effects ranging from Bloch oscillations[2] to enhanced coherent transport via controllable decoherence[3], adiabatic passage[4,5], Anderson localization[6], and many more[7]. Quantum walks in waveguide arrays[8,9] have been proposed for simulating particle statistics[10,11], boson sampling[12,13], quantum state transfer[14,15], quantum state generation[16,17], quantum search[18,19], optical transformation[20] and could implement 1 and 2-qubit gates[21,22] with the potential of implementing universal unitaries[23]. Waveguide arrays have been used to model topological band structures[24,25] and their interplay through non-Hermiticities generated by mode-selective gain and loss[26]. Exploiting these features led to new ways of realizing optical isolators[27,28],

generating and protecting quantum states[29–31], and implementing quantum circuits[32,33].

While reconfigurability is a common feature of integrated photonic devices[34], the waveguide arrays used so far have had static parameters, requiring the fabrication of one or more new, specific samples for each experiment[6]. These devices actualize either single-particle unitaries through cascaded, programmable Mach-Zehnder interferometers (MZI)[35–37] or a single-particle Hamiltonian that is determined by the detailed configuration of the waveguides. Exceptions, such as ref. 38, had some level of thermo-optical reconfigurability but without the ability to independently control the Hamiltonian parameters. Achieving such

[1]Quantum Photonics Laboratory and Centre for Quantum Computation and Communication Technology, RMIT University, Melbourne, VIC 3000, Australia. [2]ETH Zurich, Optical Nanomaterial Group, Institute for Quantum Electronics, Department of Physics, 8093 Zurich, Switzerland. [3]Centre for Quantum Computation and Communication Technology (Australian Research Council), Centre for Quantum Dynamics, Griffith University, Brisbane, QLD 4111, Australia. [4]Institute for Photonics and Quantum Sciences, SUPA, Heriot-Watt University, Edinburgh EH14 4AS, United Kingdom. [5]Institute of Physics, University of Muenster, 48149 Muenster, Germany. [6]Department of Physics, Indiana University Purdue University Indianapolis (IUPUI), Indianapolis, Indiana 46202, USA. [7]Department of Industrial Engineering, University of Trento, via Sommarive 9, 38123 Povo, Trento, Italy. [8]INFN-TIFPA, Via Sommarive 14, I-38123 Povo, Trento, Italy. [9]Qubit Pharmaceuticals, Advanced Research Department, Paris, France. ✉e-mail: yojoglek@iupui.edu; alberto.peruzzo@rmit.edu.au

control is a key step toward versatile photonic processors, routers, and simulators[20–23,39–45].

Here, we report on an electro-optically controllable lithium niobate waveguide array with up to 11 waveguides and 22 voltage control inputs. We demonstrate precise control over independent Hamiltonian terms to realize continuous-time evolutions for several thousands of Hamiltonian. We implemented the Aubry-André and Su-Schrieffer-Heeger (SSH) models to show the independent control over the diagonal and off-diagonal terms of the device Hamiltonian, respectively, and show two types of Anderson localization on the reconfigurable waveguide array (RWA). Overall, we realized more than 2500 Hamiltonians on a single device.

## Results

### Waveguide array overview and modeling

The array of continuously coupled optical waveguides is schematically shown in Fig. 1 and was fabricated using the annealed proton-exchange technique[46] on an x-cut lithium niobate substrate. Gold micro-electrodes are patterned on top of a silica buffer above the waveguides, as shown in Fig. 1b. To avoid voltage breakdown through air and glass, the electrodes must be patterned at sufficient separation. This is possible because the large cross-section of the mode in a proton-exchange waveguide leads to substantial coupling between waveguides more than 10 μm apart. To implement the control, the conventional thermo-optic effect employed in most integrated photonic circuits reported so far cannot be used since, with such a small gap between the electrodes, thermal cross-talk would be unavoidable[47]. In this device, the high electro-optic coefficient of lithium niobate allows ultra-high modulation speed with almost no cross-talk and power dissipation[48]. This assumption is based on the exceptional confinement of the electric field within the material due to the shielding effect from neighboring electrodes, thereby preventing cross-talk with other waveguides in the array. In addition to the reconfigurable section, fan-in and fan-out regions separate the waveguides to a 127 μm pitch for the coupling of light in and out of the chip by fiber arrays. The light source used in this work is a fiber-coupled polarized 808 nm laser diode. More details on the device design and fabrication can be found in "Methods".

The RWA is modeled, under first-order approximation, by the real-valued, symmetric, tridiagonal Hamiltonian[1]

$$H(\vec{v}) = \begin{bmatrix} \beta_1 & C_{1,2} & 0 & \dots & 0 & 0 \\ C_{1,2} & \beta_2 & C_{2,3} & \dots & 0 & 0 \\ 0 & C_{2,3} & \beta_3 & \dots & 0 & 0 \\ \vdots & \vdots & \vdots & \ddots & \vdots & \vdots \\ 0 & 0 & 0 & \dots & \beta_{N-1} & C_{N-1,N} \\ 0 & 0 & 0 & \dots & C_{N-1,N} & \beta_N \end{bmatrix} \quad (1)$$

where $\vec{v} = (V_1, V_2, ... V_{2N})$ is the set of voltages applied to the electrodes of the $N$ waveguides that tune the values of $C_{a,b}$, the coupling coefficient between waveguides $a$ and $b$, and $\beta_n$, the propagation coefficient of waveguide $n$. This Hermitian Hamiltonian generates a unitary transformation

$$U(\vec{v}) = e^{-iH(\vec{v})L}. \quad (2)$$

where $L$, the length of the effective coupled waveguide, determines the duration of the time evolution driven by the Hamiltonian. The parameters of $H(\vec{v})$ are controlled by voltages applied to the 22 electrodes in our device. Each waveguide (labeled from 1 to 11) has two electrodes applied, and the relationship between the electrode voltages (labeled from 1 to 22) and the parameters of the

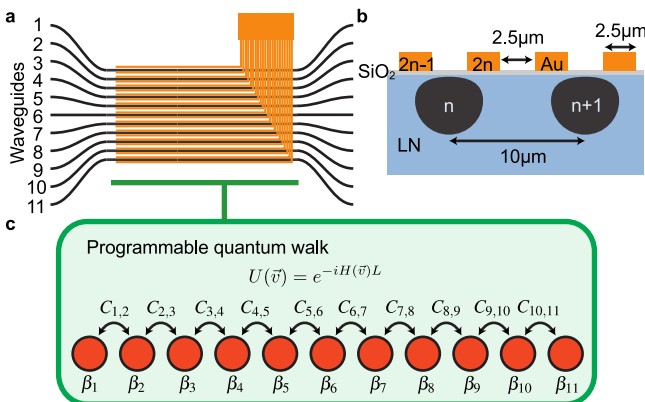

**Fig. 1 | Schematic of the 11-RWA and principle of programming Hamiltonian.** **a** The RWA has 11 waveguides (black) and 22 electrodes (orange) across the coupled region. The electrodes are controlled by multi-channel arbitrary waveform generators (Supplementary Fig. 1). **b** Cross-section of the 11-RWA device. The black regions indicate the mode shapes of the annealed proton-exchange waveguides. **c** The Hamiltonian parameters can be individually controlled with high precision.

waveguide array is given by

$$\beta_n(V) = \beta_0 + \Delta\beta_n V_{2n-1,2n} \quad (3)$$

$$C_{n,n+1}(V) = C_0 + \Delta C_1 V_{2n,2n+1} + \Delta C_2 (V_{2n-1,2n} + V_{2n+1,2n+2}) \quad (4)$$

where $n$ is the waveguide index, $\beta_0$ is the static propagation coefficient, $\Delta\beta_n$ is the voltage sensitivity of propagation coefficient and $V_{a,b} = V_a - V_b$ is the potential difference between electrodes $a$ and $b$. $C_0$ is the static coupling coefficient that encodes the bandwidth of the uniform array, $\Delta C_1$ and $\Delta C_2$ are the voltage and mode difference sensitivity of the coupling between waveguide $n$ and $n + 1$[49]. We assume $\beta_0$, $\Delta\beta_n$, $C_0$, $\Delta C_1$ and $\Delta C_2$ are consistent across the whole array in simulations. The model described by Eqs. (1)–(4) was used to fit the data of the Aubry-André and SSH experiments, as discussed in "Methods".

### Aubry-André model

The Aubry-André model describes condensed matter systems with a quasiperiodic potential that leads to a localization transition in the absence of disorder[50,51]. For a one-dimensional waveguide array, its Hamiltonian is given by a constant coupling $C_{n,n+1} = C$ and

$$\beta_n = \beta_0 + \Lambda\cos(2\pi n\chi) \quad (5)$$

where $1/\chi$ is the modulation wavelength and $\Lambda$ is the modulation amplitude. Analysis of this model predicts that when $\chi$ is an irrational (Diophantine) number, all its eigenstates become localized when the modulation amplitude exceeds the threshold $\Lambda/C = 2$[52]. In our experiments, we chose the golden mean, $\chi = (\sqrt{5}+1)/2$, meaning the modulation wavelength is $1/\chi = (\sqrt{5}-1)/2 \approx 0.618$.

From the model of our device, we have $\Lambda\cos(2\pi n\chi) = \Delta\beta_n V_{2n-1,2n}$ and $C = C_0$. To keep the coupling constant fixed, we control the electrode in pairs between adjacent waveguides (i.e., $V_{2n,2n+1} = 0$). We calculated the propagation constants change for each waveguide according to the Aubry-André model at different modulation amplitude $\Lambda$ (Fig. 2a). To ensure the voltage amplitude does not go beyond 10 V to protect the chip from damage, we set a reference voltage on the first electrode before adapting the change in the propagation constant across all waveguides by tuning the voltages across each waveguide $V_{2n-1,2n}$ according to the calculation based on the Aubry-André model. An example of the voltage setting is shown in Fig. 2b for $\Lambda/C = 37.4$.

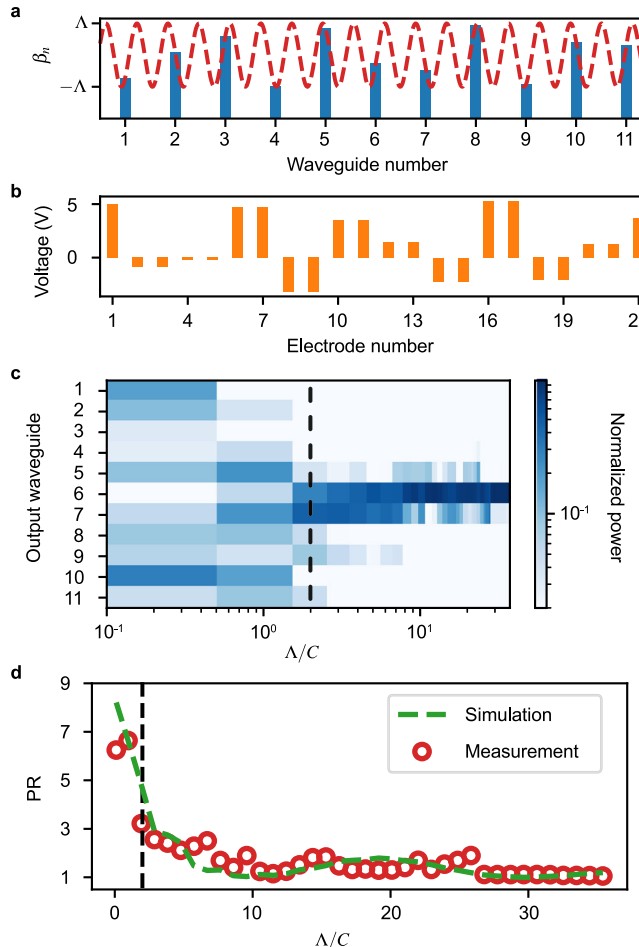

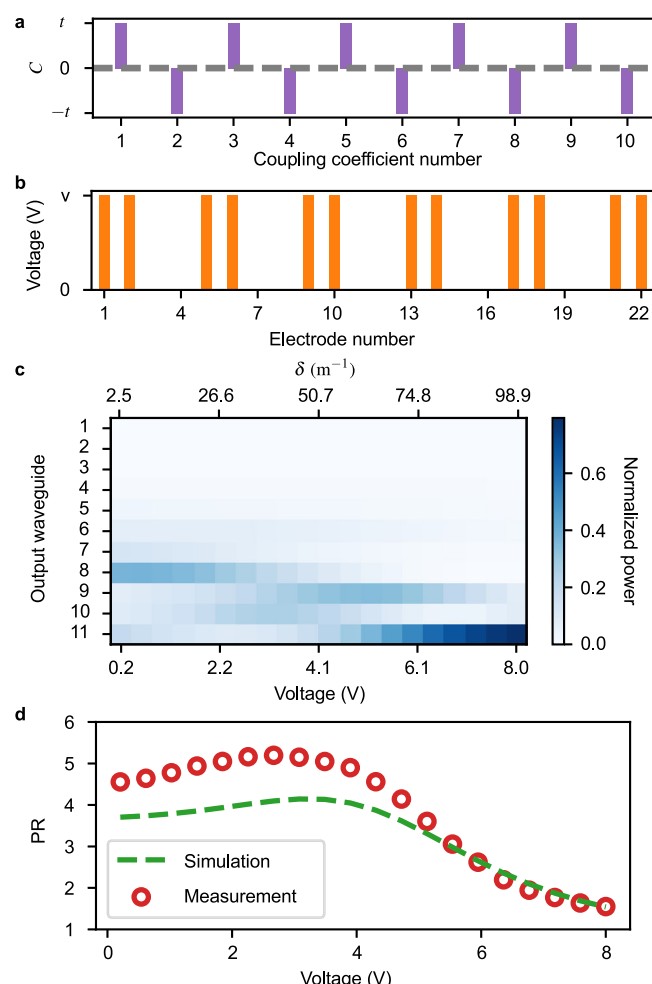

**Fig. 2 | Implementation of the Aubry-André model in an $N = 11$ array. a** The propagation constants are modulated based on Eq. (5) where $\beta_n$ are offset from $\beta_0$. The red dashed line shows the quasiperiodic potential profile with wavelength $1/\chi \approx 0.618$. **b** A sample of the voltage configuration used to implement the modulation is shown in (**a**), corresponding to $\Lambda/C = 37.4$. **c** Output power distribution as a function of potential strength $\Lambda/C$. Each vertical slice of the 38 shown here is the measurement for a different Hamiltonian, actualized via voltage control. The black dashed line indicates the phase transition at $\Lambda/C = 2$. **d** Measured (and simulated) participation ratio (PR) showing the transition to a localized regime.

**Fig. 3 | Implementation of the SSH model. a** The coupling constants are modulated based on Eq. (6), where $t$ is the offset from $C_0$ (gray dashed line). **b** Voltages are applied across electrodes within each unit cell, alternating $V$ to 0 V to implement the coupling constants modulation in (**a**). **c** We plot the output power distribution as a function of the dimerization strength. Each vertical slice of the 20 presented is the measurement with a different Hamiltonian, with the corresponding $\delta$ shown. **d** Measured (and simulated) PR shows the edge state localized at waveguide 11.

We injected light in the middle waveguide and measured the power at the output facet of the waveguide array for different potential strengths $\Lambda/C$, which is shown in Fig. 2c. As predicted by the Aubrey-André model, the sharp localization transition is observed around $\Lambda/C = 2$. We use the participation ratio (PR) to quantify the degree of localization[53]. It is given by $PR = 1/\sum_i |P_i|^2$, where $P_i$ is the normalized output power at waveguide $i$, i.e., $\sum_i P_i = 1$. The measured (and simulated) PR, Fig. 2d, changes from $PR \sim N$, indicating extended states, to $PR \sim 1$, indicating all localized states. The fidelity with the theoretical model is $0.949 \pm 0.009$ (Supplementary Fig. 3).

## Su-Schrieffer-Heeger model

>The SSH model, first used to model soliton formation in *trans*-poly-acetylene, is a minimal model with a topologically nontrivial band structure. Its dynamics can be described by the Hamiltonian[54] with $\beta_n = \beta_0$ and

$$C_{2n-1,2n} = T_1,$$
$$C_{2n,2n+1} = T_2. \quad (6)$$

$T_1$ and $T_2$ denote the intra-cell and inter-cell coupling coefficients, where unit cells are formed by A and B sublattices consisting of odd- and even-numbered waveguides, respectively. Our device, with 11 waveguides, is terminated at one end with an extra A waveguide and has a zero-energy edge state that is localized near the last waveguide when $T_1 > T_2$. For the experiments performed with our device, we have $T_1 = C_0 + t$ and $T_2 = C_0 - t$, as shown in Fig. 3a, where the dimerization strength $\delta = |T_1 - T_2| = |2t|$ is limited by the maximum voltage amplitude we applied to the electrodes.

The propagation constant is fixed by connecting two electrodes on top of each waveguide (i.e., $V_{2n-1,2n} = 0$). We applied voltages to electrodes across the two waveguides within each cell, which can modulate the strength of dimerization $\delta$ by tuning the voltage amplitude, as shown in Fig. 3b. We injected light in the last waveguide and measured the power distribution at the output facet of the waveguide array (Fig. 3c) at each dimerization strength. The edge state can be clearly observed when the applied voltage amplitude is around 6.1 V ($\delta = \gamma * V = 75.6 \text{ m}^{-1}$). The dimerization coefficient $\gamma(V) \approx 12.4 \text{ m}^{-1}\text{V}^{-1}$ is given by our simulation. The fidelity with the theoretical model is $0.904 \pm 0.001$ (Supplementary Fig. 3). In Fig. 3d, we report the

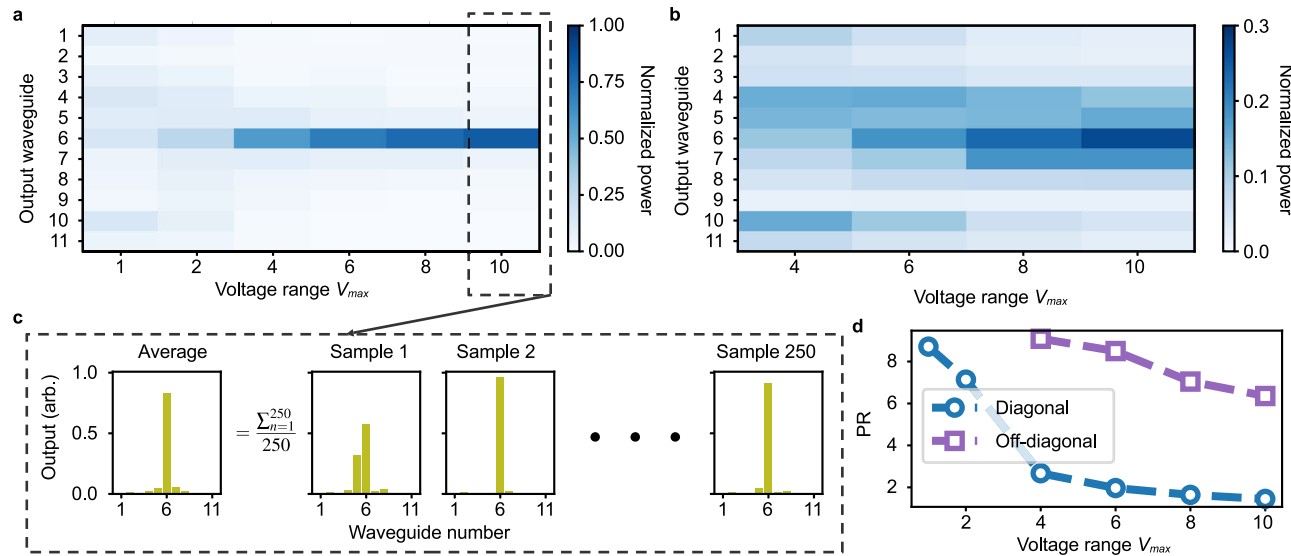

**Fig. 4 | Anderson localization with site and bond disorder.** Each vertical slice is the averaged power distribution measured at the output facet corresponding to a disorder strength indicated by the voltage range. **a** The disorder is applied to the propagation coefficients $\beta_n$. **b** The disorder is applied to the propagation coefficients $C_{n,n+1}$. **c** Each averaged power distribution comes from 250 random Hamiltonians. **d** Measured PR of Anderson localization.

simulated and measured PR, which also shows the emergence of an edge state localized near the last waveguide.

## Anderson localization

In 1958 Anderson showed that in low dimensions, arbitrary disorder in a crystal localizes any wavefunction[55], thereby elucidating a universal property of disordered wave systems. We can simulate these dynamics in a waveguide array starting from the Hamiltonian in Eqs. (1) with voltage-controlled propagation and coupling coefficients according to Eqs. (3) and (4). The eigenmodes of the Hamiltonian become more localized with the increase of the disorder induced in the lattice. To simulate these dynamics, we performed two sets of measurements, one randomizing only the propagation constants $\beta_n$ with $C_{n,n+1} = C_0$ and one randomizing the coupling constants $C_{n,n+1}$ while keeping $\beta_n = \beta_0$ for every $n$ (diagonal disorder and off-diagonal disorder). For both cases, light is launched into the central waveguide, corresponding to waveguide number 6. The values of $\beta_n$ or $C_{n,n+1}$ are fixed by short-circuiting the corresponding electrodes, i.e., $V_{2n-1,2n} = 0$ or $V_{2n,2n+1} = 0$, respectively (Supplementary Fig. 4).

The strength of the disorder depends on the maximum voltage $V_{max}$ used to change the values of $\beta_n$ and $C_{n,n+1}$. The dimensionless measure of the disorder can be defined as the ratio of the disorder modulation to the zero-disorder bandwidth, i.e., $\Delta\beta V_{max}/C_0$ or $\Delta C V_{max}/C_0$[56]. To carry out the averaging over different, disordered Hamiltonians, we create 250 samples (Fig. 4c) for each disorder strength with uniformly randomized voltage in the range of $-V_{max}$ to $+V_{max}$ and plot the average output power distributions of those samples with the input light in the middle waveguide in Fig. 4a, b. Hence, this experiment corresponds to a total of 2500 static devices. The estimated diagonal and off-diagonal disorder strengths in the chip, given by fitting the measurements, are $\Delta\beta/C_0 \approx 3.4 V^{-1}$ and $\Delta C/C_0 \approx 0.07 V^{-1}$, respectively. The corresponding PR as a function of maximum voltage is shown in Fig. 4d. Since the off-diagonal disorder is fifty times weaker than the diagonal disorder, the reduction in the PR is correspondingly smaller.

## Discussion

Reconfigurable waveguide arrays are a powerful tool for optical signal routing, implementing time-evolution Hamiltonians to simulate quantum materials and phenomena, and can be cascaded[57] to realize arbitrary unitaries. The use of the electro-optic effect for controlling the Hamiltonian coefficients offers key advantages in terms of speed and suppressed cross-talk when compared to thermal shifters. By simulating the Aubry-André, SSH, and Anderson Hamiltonians, we have shown how one reconfigurable device can replace the fabrication of thousands of static devices with a high level of control of the independent parameters.

In our experiments, we limited the maximum voltage amplitude to 10 V to protect the device from damage. However, as seen in Fig. 4b, the amount of off-diagonal disorder it induced is insufficient to observe strong localization. This can be improved by optimizing the electrode design (such as decreasing the distance between electrodes).

The accuracy of the theoretical model used for fitting the data is limited by the tridiagonal real-valued Hamiltonian approximation. Many experimental factors also affect the model fidelity. Firstly, fabrication errors in the waveguides, such as the waveguide width and separation, can cause $\beta_0$ and $C_0$ to be not identical across the array. Secondly, the dimensions, position, and quality of the fabricated electrodes can vary, making $\Delta\beta_n$ and $\Delta C_{n,n+1}$ site-dependent. This could explain the difference in fidelities between the Aubry-André and SSH experiments. Since the two experiments rely on different electrode configurations, deviations such as misalignment of electrodes mask and non-uniform electrode resistance might cause different performance and therefore fidelity. To improve the model fidelity, one can utilize machine-learning-based solutions proposed and demonstrated in[58,59] and implement more accurate identification and control of the device.

Additionally, using more advanced fabrication methods, particularly for the electrodes, will also help improve performance and scalability. In particular, the z-cut thin film lithium niobate platform may allow reducing the footprint–enabling the fabrication of cascaded RWAs–while keeping the advantage of electro-optic performance, including spacing the electrodes, high-speed operations and low driving voltage[60,61]. Moreover, it offers enhanced nonlinearity[62], the possibility of integrating single-photon sources[63,64] and detectors[65] on chips, cryogenic compatible operations, increasing the scalability and reducing the coupling losses.

Increasing scalability, performance and efficiency are the prime focus in advanced integrated photonic systems. In theory, thousands

of waveguides at a coupling distance of 10 μm can be patterned on a 6-inch lithium niobate wafer. Even with the electrical wiring and fan-in fan-out of the waveguide to match fiber arrays, it is feasible to produce a device with hundreds of waveguides. Despite the capability of femtosecond-laser-written waveguides[66] to implement arbitrary or time-dependent Hamiltonians, they are generally limited to static devices. Furthermore, although large circuits incorporating MZIs as fundamental reconfigurable units have been extensively demonstrated[36,37], the scalability of MZIs is impeded by their sensitivity to fabrication errors and their susceptibility to significant bending losses[67,68]. Compared with the scheme by Clements et al.[69], the waveguide array-based scheme[23] incurs lower bending losses due to the half number of bending sections the photons will experience.

The wave equation in the paraxial and scalar approximations is similar to the Schrodinger equation, where the refractive index profile plays the role of a potential. This is why waveguide array structures have been used to simulate solid-state quantum phenomena with laser light[10]. Furthermore, laser light characterization of the Hamiltonian completely predicts the quantum process of linear photonic devices[70,71], and a small spectral difference between laser light and single-photon sources will give a negligible change in the device Hamiltonian. Therefore, with the ability to independently and accurately control the propagation and coupling parameters of a waveguide array, this new structure can be used for a variety of applications, ranging from simulating complex physics systems[41–43] to processing quantum information[20,22,23,44,45] as well as continuous simulation of time for complex time-evolution dynamics of arbitrary Hamiltonians.

## Methods
### Details of the device
The device is fabricated using annealed and reverse proton-exchange technology with x-cut bulk lithium niobate[46,59,72]. The length of the continuously coupled region of the device is 24 mm. The propagation constant $\beta_0$ is given by

$$\beta_0 = \frac{2\pi}{\lambda} n_0 \qquad (7)$$

where $n_0 = 2.1753$ is the effective refractive index of the designed waveguide, $\lambda$ is the light wavelength. We used the model described in the main text to fit the static chip characterization data and the Aubry-André and SSH model measurements. The fitting parameters after optimization are $\Delta\beta(V) = 290.6 \ m^{-1}V^{-1}$, $C_0 = 84.8 \ m^{-1}$, $\Delta C_1(V) = 6.2 \ m^{-1}V^{-1}$ and $\Delta C_2(V) = -8.5 \ m^{-1}V^{-1}$. In the fitting, we only fit the model with real values. Hence we assumed $C_{n,n+1} = C_{n+1,n}$.

### Device measurement and control
A schematic of the experimental setup is shown in Supplementary Fig. 1. A polarized 808 nm laser and multi-channel fiber-coupled high-speed optical power meter were used for the output measurements. Polarization-maintaining fiber (PMF) arrays with 127 um pitch were used for butt-coupling to the chip at the input and output sides with a coupling loss of 4.9 dB per facet (68%), caused by fiber-waveguide mode mismatch. A total power loss, including the fiber-to-chip coupling and propagation loss, of 9.8 dB (90%) was measured. This is sufficient for two-photon quantum experiments, as demonstrated in the first two-photon quantum walk in a static waveguide array[9]. To enhance the total transmission of the device, for example, for experiments using more than two photons, the insertion loss can be reduced by various technologies, ranging from on-chip components to engineered fibers, such as high-index fibers that reduce the mode field diameter of the fiber[73–77]. Additionally, losses can be mitigated by improving the fabrication process and/or working at telecom wavelength where propagation and coupling losses are lower[2,46].

Lithium niobate suffers from electric charges accumulating in the $SiO_2$ buffer layer under the control electrodes, which results in a drift of the optical output when a voltage is applied. To mitigate the output optical drift, 1.66 s non-biased square pulses were applied in order to achieve unbiased control, i.e., each target voltage is followed by a pulse of the same magnitude and opposite sign (Supplementary Fig. 2a). Each electrode was connected to an independent output channel of an arbitrary waveform generator (AWG), and multiple AWGs were synchronized with an external trigger. To reset the chip state, a 20 s gap between each measurement is applied[78]. In Supplementary Fig. 2b, we report the chip response of the pulsing scheme used to reset the chip to the initial state (where the Hamiltonian is not affected by the voltage). The data for each measurement is taken from the average of the output power in the first half of the non-biased square pulse after the rising time (-0.22 s), which is limited by the built-in low-pass filter of the multi-channel power meter. We characterized the rise time as less than 0.2 ms based on optical response with fast photodiodes, and optical outputs keep drifting due to electric charging after the rise time. The measurements can be sped up by using modulated square pulses with a higher frequency[79].

### Simulation of the voltage-controlled output distribution for the Aubry-André and SSH Hamiltonians
The simulations shown in Supplementary Fig. 3 are performed by numerical optimization based on the theoretical model of the device and assumptions described in the main text. The fidelity between simulated and measured output distributions is calculated as $F = \frac{1}{N}\sum_{n=1}^{N} F_n$, where $N$ is the total number of measurements, $F_n = \sum_i \sqrt{P_i^S P_i^M}$ is the fidelity for each prepared output distribution and $P_i^S(P_i^M)$ is the normalized simulated (measured) power distribution at the output waveguide i. The fidelity for the Aubry-André and SSH models is $0.949 \pm 0.009$ and $0.904 \pm 0.001$, respectively. The static Hamiltonians fitting fidelity is $0.878 \pm 0.001$ with parameters fitted from Aubry-André and SSH model measurements.

### Anderson localization experiments controlling scheme
In Supplementary Fig. 4, we show how we controlled the Hamiltonian parameters through electrodes. The pink dashed line indicates $\beta_0$ and $C_0$ as zero-disorder level. The bars in Supplementary Fig. 4a, c indicate a randomized change in the Hamiltonian parameters. The voltage amplitude of every electrode is randomized in a limited range to realize different disorder levels. A larger voltage range indicates a higher disorder level. The maximum voltage range we set is from −10 V to 10 V to protect the device.

## Data availability
The authors declare that the fitting parameters and data supporting the findings of this study are available within the paper and its Supplementary Information. The raw data in this study have been deposited in the Figshare database under the accession code https://doi.org/10.6084/m9.figshare.24587775. They are also available from the corresponding author upon request.

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

## Acknowledgements

A.P. acknowledges an RMIT University Vice-Chancellor's Senior Research Fellowship and a Google Faculty Research Award. M.L. was supported by the Australian Research Council (ARC) Future Fellowship (FT180100055). B.H. was supported by the Griffith University Post-doctoral Fellowship. Y.N.J. was supported by ONR Grant No. N00014-21-1-2630. This work was supported by the Australian Government through the Australian Research Council under the Centre of Excellence scheme (No: CE170100012) and the Griffith University Research Infrastructure Program. This work was partly performed at the Queensland node of the Australian National Fabrication Facility, a company established under the National Collaborative Research Infrastructure Strategy to provide nano- and microfabrication facilities for Australia's researchers.

## Author contributions

Y.Y., R.J.C., and A.P. conceived and designed the project. Y.N.J., M.L., and A.P. directed the project. Y.Y. carried out the simulations and experiments. B.H., F.L., and M.L. fabricated the device. Y.Y., Y.N.J., and A.P. prepared the manuscript. Y.Y., R.J.C., Y.N.J., M.L., and A.P. analyzed the data and discussed the results.

## Competing interests

The authors declare no competing interests.
