## [Peer Review File · Nature Communications]

Programmable high-dimensional Hamiltonian in a photonic waveguide arrayREVIEWER COMMENTS

Reviewer #1 (Remarks to the Author):

The present manuscript reports on a programmable photonic processor based on a waveguide array operating in the continuous coupling regime. More specifically, the platform is based on a lithium niobate substrate, where the circuit operation can be controlled via the electro-optic effect. At variance with previous implementations, this approach permits to singularly reconfigure the propagation constant of the waveguide and the coupling coefficient between adjacent waveguides. The authors fabricated an 11-mode, 22-reconfigurable elements devices, and show the capability to successfully program its operation to implement different Hamiltonian dynamics.

As an overall assessment, this paper represents an interesting implementation. In particular, the capability of addressing and controlling the operation of the single waveguides in a reconfigurable way provides a significant step forward in the adoption of waveguide arrays, based on the continuous coupling approach, as a programmable photonic processor.

Starting from this assessment, I have a few comments on the manuscript before I can fully support its publication in Nature Communications.

(1) An important aspect for the operation of reconfigurable circuits regards the calibration of the active elements response. In this case, this corresponds to characterizing the transfer function (in other words, the map) between the applied voltages and the actual change in the parameters. In the present system, this is expressed by Eqs. (3-4), which requires to characterize a few parameters (whose values are reported in the Supplementary Materials). I have a couple of comments on this: -) On one side, it would be nice to have a more detailed discussion in the Supplementary Materials, with corresponding data and procedure, from where the parameters ($\Delta \beta$, C_0 , ...) to be used in Eqs. (3-4) were obtained. -) As a second aspect, one of the key points, that strongly simplifies the characterization procedure, is the negligible effect of cross-talks. In the calibration process, was this absence of cross-talk also tested/verified from the data, or was it assumed given the architecture and the fabrication process of the device? Adding this discussion would benefit to provide a complete characterization of the platform.

(2) At a first glance, a better agreement is found between the experimental data and the simulation for the Aubry-Andre model rather than for the SSH model. In the first case, the parameters are modulated by changing only the propagation constant β , while in the second case the dynamics is obtained by fixing the propagation constant and by modulating the coupling coefficients C . This corresponds to a different pattern of voltages which needs to be applied on the device.

Does this correspond to a reduced control in the coupling coefficients rather than on the propagation constants? I would suggest the authors to add a discussion on this aspect.

Reviewer #2 (Remarks to the Author):

The Authors report on a novel photonic device, consisting in a linear array of 11 evanescently-coupled optical waveguides, fabricated in lithium niobate substrate and reconfigurable via the electro-optic effect. In detail, the electro-optic reconfiguration is operated by means of 22 electrodes running above the array, parallel to the waveguides for their whole length. Each waveguide is provided of two electrodes, deposited above its right side and above its left side. By applying a voltage to two electrodes that comprise a given waveguide, its propagation constant

can be finely tuned. By applying a voltage to electrodes placed between two adjacent waveguides, it is possible to tune their coupling coefficient. The potentials of this device in realizing different Hamiltonians are tested with several experiments. In particular, the manuscript reports implementations of the Aubry-André model, of the SSH model and of different static disorder configurations that realize Anderson localization.

The manuscript is well organized and well written, and the proposed device shows surely some points of technological novelty. However I am not sure that this work represents, in the fields of integrated photonics and optical analog simulations, an advance of such importance to worth publication in Nature Communications. My main question is whether this device enables to simulate new physical phenomena or give access to relevant experimental results hard to achieve with other platforms.

Optical simulations based on optical waveguide arrays have flourished in the last couple of decades. The Authors report indeed in the introduction of their manuscript a number of references that testify the wide range of physical phenomena that have been experimentally observed, in the past years, using waveguide array platforms. One can note in particular that, by using the femtosecond laser direct writing technique, it was possible to produce high quality waveguide arrays implementing diverse Hamiltonian evolutions (see e.g. A. Szameit & S. Nolte, *J. Phys. B: At. Mol. Opt. Phys.* 43, 163001 (2010); and S. Longhi, *Laser Photon. Rev.* 3, 243-261 (2009)), and within quick fabrication times. In addition, that technique enabled the experimental realization of time-dependent Hamiltonians (i.e. modulation of the waveguide properties along the propagation coordinate) with basically no added effort. It further has to be noted that, nowadays, universal photonic processors are demonstrated with up to 20 modes and more (see e.g. Taballione et al. arXiv preprint, arXiv:2203.01801). Such devices reproduce totally arbitrary linear transformations of optical modes (i.e. arbitrary Hamiltonians acting for an arbitrary long time), using thermo-optic phase shifters, and with fidelities robustly above 95%. Finally, less universal processors have been demonstrated with an even larger number of optical modes: a term of comparison for the present device could be the programmable nanophotonic processor demonstrated by Harris et al. (*Nat. Photon.* 11, 447-452 (2017)), which was used to demonstrate experimentally quantum transport dynamics in up to 26 modes, and with fidelities above 99% with respect to theory.

I think that the above considerations are important to provide a fair context to the present device. As a matter of fact, here we have a 11-waveguide device that can realize only a limited set of Hamiltonian evolutions, i.e. those governed by the tridiagonal matrix given by Eq. (1), and only for a limited range of values of the coefficients. These Hamiltonians are a small subset of those accessible in static devices fabricated with very low cost by femtosecond laser writing. Current universal photonic processors would surely be able to implement these Hamiltonians, and they would also be fundamentally accessible to the non-universal processor demonstrated by Harris et al.

In comparison, it seems that the only relevant feature of the present device is the far higher reconfiguration speed. However, I am not sure that this feature, while technically substantiated, can give a real, game-changing advantage in a significant range of applications in the field of optical simulations.

If a limited number of realizations are needed, static devices realized by femtosecond laser writing may be still a powerful choice. Given typical waveguide inscription speeds of tens of mm/s (see e.g. Ref. 14) one can estimate that a single 11-waveguide array would take about 1 minute to be realized; one experiment (including the characterization steps) would be accomplished in a few hours time. Note that this platform gives ample freedom in the choice of the operating wavelength and of the array geometry and parameters. If instead we consider the possibilities provided by a universal photonic processor such as the ones reported in the literature, based on the thermo-optic effect, we note that reconfigured can be accomplished in less than 1 second time. Thus, scanning the ~ 2500 realizations shown in this manuscript might take less than 1 hour, not such a long time on the scales of scientific experiments. An even shorter time would be required if a thermo-optic nanophotonic device such as the one demonstrated by Harris et al. was used. Hence, it seems that one would really need the present device only if there is the requirement of a huge

number of Hamiltonian realizations which, by chance, happens to be among the few ones that are at reach of its non-universal layout.

The authors also mention the suppression of cross-talks among the potentials of their device. This is expected to guarantee high fidelities. However, I observe that the measured fidelities in reaching the theoretical distributions at the output are not exceptionally high, especially if compared to the mentioned works by Taballione et al. or by Harris et al.

On this aspect one can also note the non-negligible discrepancy in the measured/simulated participation ratios (PR) in Figure 3d. Since the PR is a collective figure of merit that should be quite robust to imperfections, it may be an indication that control on the waveguide parameters is only obtained with some approximation.

In the end, I think it cannot be concluded that this electro-optic device provides outstanding applicative advantages with respect to the other photonic processors reported in the literature. I would not recommend publication of this manuscript in Nature Communications, but rather I think this work would deserve to be showcased in a more specialized journal, where the photonic-technology aspects would likely find a solid appreciation.

In addition, I would strongly suggest the authors to address the following points, which in my opinion are essential for a fair assesment of the quality of the reported results:

- 1) the operating wavelength should be clearly stated;
- 2) characterization of the device insertion loss should be reported and discussed; if possible, an estimate of the different loss contributions should also be given (e.g. input/output coupling loss due to mode matching, propagation loss per unit length of the waveguide...)
- 3) the authors should discuss with some detail how they have calibrated precisely the action of each electrode, namely how they have estimated the coefficients $\Delta\beta_n$, ΔC_1 and ΔC_2 in Eq. (3) and (4).
- 4) the uniformity of the static and reconfigurable waveguide parameters along the propagation coordinate should be discussed or analyzed. For instance, the authors might implement a nominally uniform Hamiltonian, measuring the amount of "noise" in the output distribution and fitting it back to simulated models. Other more appropriate ways could anyway be devised and adopted.
- 5) the time-response of the device should be characterized experimentally.

Reviewer #3 (Remarks to the Author):

In this work, Yang et al., demonstrate a reconfigurable waveguide array fabricated in an annealed proton exchange lithium-niobate platform. The authors exploit the reconfigurability of their system to implement different input/output transformations corresponding to different Hamiltonians. Specifically, they consider the Su-Schrieffer-Heeger (SSH) model, the Aubrey-Andre model, and Anderson localization. The linear characterization of the transmission function of the structure shows a reliable and effective reconfigurability of their structure. Fidelities between the expected and measured transmission function are above 90% for all the examined configurations. The main claim of the manuscript is that this platform can find applications for quantum information processing in high-dimensional systems.

These experimental results are solid and convincing to demonstrate that their platform is flexible and reconfigurable in a predictive way. However, the claim that this platform will have an impact on quantum information processing is not demonstrated. First, all the experimental results presented in the manuscript are obtained with classical light. Second, it is not clear what are the limits of the coupled-waveguide approach in terms of the Hamiltonian can be realized. It seems as if the authors chose SSH and Aubrey-Andre models because they are the easiest to be studied in coupled waveguide arrays, however, their relevance for quantum information processing is not

clear. Third, the scalability of the approach is not clear. This ultimately depends on the geometrical configuration (coupled waveguide array), the refractive index contrast, and the maximum length of the system. How many waveguides one can realistically implement? Other details regarding possible issues, such as DC drift in LiNbO₃, are not discussed. These are crucial aspects given the ambitious claims of the authors.

The manuscript is sometimes a bit misleading. Indeed, in the introduction, the authors cite several papers presenting platforms in which quantum experiments have been carried out. The conclusions also point in the direction of quantum information processing. However, all the experimental results are obtained using classical light, to which the authors often refer as "input light", without specifying or stressing that it is classical in nature. While such a linear characterization is important, it is not sufficient to establish what the performances with nonclassical light would be. The authors discuss advantages such as the possibility of integrating single-photon sources and detectors on chips. However, these are not demonstrated in the present system.

Given the numerous experimental demonstrations of quantum light manipulation in complex photonic integrated circuits in various material platforms, I am afraid the results reported here are not significant enough to grant publication in Nature Communication. I believe that the manuscript is more suitable for other kinds of journals, e.g., Applied Physics Letter or Optics Express.

Referee 1

Comment: *The present manuscript reports on a programmable photonic processor based on a waveguide array operating in the continuous coupling regime. More specifically, the platform is based on a lithium niobate substrate, where the circuit operation can be controlled via the electro-optic effect. At variance with previous implementations, this approach permits to singularly reconfigure the propagation constant of the waveguide and the coupling coefficient between adjacent waveguides. The authors fabricated an 11-mode, 22-reconfigurable elements devices, and show the capability to successfully program its operation to implement different Hamiltonian dynamics.*

As an overall assessment, this paper represents an interesting implementation. In particular, the capability of addressing and controlling the operation of the single waveguides in a reconfigurable way provides a significant step forward in the adoption of waveguide arrays, based on the continuous coupling approach, as a programmable photonic processor.

Starting from this assessment, I have a few comments on the manuscript before I can fully support its publication in Nature Communications.

Response: We thank the reviewer for their time and effort in reviewing the manuscript and providing useful comments and feedback.

Comment: *(1) An important aspect for the operation of reconfigurable circuits regards the calibration of the active elements response. In this case, this corresponds to characterizing the transfer function (in other words, the map) between the applied voltages and the actual change in the parameters. In the present system, this is expressed by Eqs. (3-4), which requires to characterize a few parameters (whose values are reported in the Supplementary Materials). I have a couple of comments on this: -) On one side, it would be nice to have a more detailed discussion in the Supplementary Materials, with corresponding data and procedure, from where the parameters ($\Delta\beta$, C_0 , ...) to be used in Eqs. (3-4) were obtained.*

Response: The Hamiltonian coefficients were calculated by fitting the data measured for the SSH and Aubry-André experiments to the models in Eqs (1-4), as discussed in the Supplementary Methods 'Design details of the device' section. To further clarify this, we have added the following sentence in the main text under the section 'Waveguide array overview and modeling':

The model described by Eqs (1-4) was used to fit the data of the Aubry-André and SSH experiments, as discussed in Supplementary Methods.

The high fidelity of the fit verified the validity of the model.

Comment: *-) As a second aspect, one of the key points, that strongly simplifies the characterization procedure, is the negligible effect of cross-talks. In the calibration process, was this absence of cross-talk also tested/verified from the data, or was it assumed given the architecture and the fabrication process of the device? Adding this discussion would benefit to provide a complete characterization of the platform.*

Response:

The reviewer is correct that there's an underlying assumption of negligible cross-talk, which is incorporated in the definition of the Hamiltonian. This assumption is based on the exceptional confinement of the electric field within the material due to the shielding effect from neighboring electrodes, thereby preventing cross-talk with other waveguides in the array. This was demonstrated, for example, in a recent LN-based study [1], where it was shown that the thermo-optic effect propagates in a diffusive way along the material, while the cross-talk for electro-optic devices is absent. Even with more advanced fabrication techniques such as isolation trenches and bridge waveguide [2], the thermal cross-talk still affects neighboring devices in a distance of hundreds of μm . The thermal cross-talk is a significant drawback in commonly used photonic platforms such as silica, silicon, or silicon nitride.

We added references [1, 2] in the following text in the 'Waveguide array overview and modeling' section:

To implement the control, the conventional thermo-optic effect employed in most integrated photonic circuits reported so far, cannot be used since with such a small gap between the electrodes, thermal cross-talk would be unavoidable [2]. In this device, the high electro-optic coefficient of

lithium niobate allows ultra-high modulation speed with almost no cross-talk and power dissipation [1]. This assumption is based on the exceptional confinement of the electric field within the material due to the shielding effect from neighboring electrodes, thereby preventing cross-talk with other waveguides in the array.

Comment: (2) *At a first glance, a better agreement is found between the experimental data and the simulation for the Aubry-Andre model rather than for the SSH model. In the first case, the parameters are modulated by changing only the propagation constant β , while in the second case the dynamics is obtained by fixing the propagation constant and by modulating the coupling coefficients C . This corresponds to a different pattern of voltages which needs to be applied on the device. Does this corresponds to a reduced control in the coupling coefficients rather than on the propagation constants? I would suggest the authors to add a discussion on this aspect.*

Response: As the reviewer suggested, the electrodes might not all perfectly behave as modeled. Since the two experiments rely on different electrode configurations, deviations such as misalignment of electrodes mask and non-uniform electrode resistance might cause different performance and therefore fidelity.

We added the following sentence to discuss the fidelities in the 'discussion' section:

This could explain the difference in fidelities between the Aubry-André and SSH experiments. Since the two experiments rely on different electrode configurations, deviations such as misalignment of electrodes mask and non-uniform electrode resistance might cause different performance and therefore fidelity.

Referee 2

Comment: *The Authors report on a novel photonic device, consisting in a linear array of 11 evanescently-coupled optical waveguides, fabricated in lithium niobate substrate and reconfigurable via the electro-optic effect. In detail, the electro-optic reconfiguration is operated by means of 22 electrodes running above the array, parallel to the waveguides for their whole length. Each waveguide is provided of two electrodes, deposited above its right side and above its left side. By applying a voltage to two electrodes that comprise a given waveguide, its propagation constant can be finely tuned. By applying a voltage to electrodes placed between two adjacent waveguides, it is possible to tune their coupling coefficient. The potentials of this device in realizing different Hamiltonians are tested with several experiments. In particular, the manuscript reports implementations of the Aubry-André model, of the SSH model and of different static disorder configurations that realize Anderson localization.*

The manuscript is well organized and well written, and the proposed device shows surely some points of technological novelty. However I am not sure that this work represents, in the fields of integrated photonics and optical analog simulations, an advance of such importance to worth publication in Nature Communications. My main question is whether this device enables to simulate new physical phenomena or give access to relevant experimental results hard to achieve with other platforms.

Response: We thank the referee for their time and effort in reviewing our manuscript and providing valuable feedback.

Comment: *Optical simulations based on optical waveguide arrays have flourished in the last couple of decades. The Authors report indeed in the introduction of their manuscript a number of references that testify the wide range of physical phenomena that have been experimentally observed, in the past years, using waveguide array platforms. One can note in particular that, by using the femtosecond laser direct writing technique, it was possible to produce high quality waveguide arrays implementing diverse Hamiltonian evolutions (see e.g. A. Szameit & S. Nolte, J. Phys. B: At. Mol. Opt. Phys. 43, 163001 (2010); and S. Longhi, Laser Photon. Rev. 3, 243-261 (2009)), and within quick fabrication times. In addition, that technique enabled the experimental realization of time-dependent Hamiltonians (i.e. modulation of the waveguide properties along the propagation coordinate) with basically no added effort. It further has to be noted that, nowadays, universal photonic processors are demonstrated with up to 20 modes and more (see e.g. Taballione et al. arXiv preprint, arXiv:2203.01801). Such devices reproduce totally arbitrary linear transformations of optical modes (i.e. arbitrary Hamiltonians acting for an arbitrary long time), using thermo-optic phase shifters, and with fidelities robustly above 95%. Finally, less universal processors have been demonstrated with an even larger number of optical modes: a term of comparison for the present device could be the programmable nanophotonic processor demonstrated by Harris et al. (Nat. Photon. 11, 447-452 (2017)), which was used to demonstrate experimentally quantum transport dynamics in up to 26 modes, and with fidelities above 99% with respect to theory.*

I think that the above considerations are important to provide a fair context to the present device. As a matter of fact, here we have a 11-waveguide device that can realize only a limited set of Hamiltonian evolutions, i.e. those governed by the tridiagonal matrix given by Eq. (1), and only for a limited range of values of the coefficients. These Hamiltonians are a small subset of those accessible in static devices fabricated with very low cost by femtosecond laser writing. Current universal photonic processors would surely be able to implement these Hamiltonians, and they would also be fundamentally accessible to the non-universal processor demonstrated by Harris et al.

In comparison, it seems that the only relevant feature of the present device is the far higher reconfiguration speed. However, I am not sure that this feature, while technically substantiated, can give a real, game-changing advantage in a significant range of applications in the field of optical simulations.

If a limited number of realizations are needed, static devices realized by femtosecond laser writing may be still a powerful choice. Given typical waveguide inscription speeds of tens of mm/s (see e.g. Ref. 14) one can estimate that a single 11-waveguide array would take about 1 minute to be realized; one experiment (including the characterization steps) would be accomplished in a few hours time. Note that this platform gives ample freedom in the choice of the operating wavelength and of the array geometry and parameters. If instead we consider the possibilities provided by a universal photonic processor such as the ones reported in the literature, based on the thermo-optic effect, we note that reconfigured can be accomplished in less than 1 second time. Thus, scanning the 2500 realizations shown in this manuscript might take less than 1 hour, not such a long time on the scales of scientific experiments. An even shorter time would be required if a thermo-optic nanophotonic device such as the one demonstrated by Harris et al. was used. Hence, it seems that one would really need the

present device only if there is the requirement of a huge number of Hamiltonian realizations which, by chance, happens to be among the few ones that are at reach of its non-universal layout.

Response: The femtosecond laser writing technology has supported the studies of many optical simulations of physical phenomena. However, a reconfigurable approach is fundamentally different and more flexible than fabricating fixed devices. Firstly, each fabricated device needs to be characterized, which is time-consuming as each device will behave differently. Once we have a model for the reconfigurable device, we can dial any Hamiltonian that this device can implement. Instead of having a Hamiltonian for each device, the reconfigurable has a unique one. Having many static devices also incurs extra alignment work. To date, large-scale reconfigurable devices have only been demonstrated for discrete circuits not continuously coupled waveguide arrays.

In addition, the thermo-optic effect propagates in a diffusive way along the material and can lead to low-fidelity circuits if the micro-heaters are not well-separated [1, 2], which is a significant drawback in commonly used photonic platforms such as silica (commonly used in femtosecond laser direct writing), silicon, or silicon nitride (commonly used in discrete photonic networks).

In this work, we have fabricated and demonstrated the most complex electro-optic device reported to date, which can potentially be operated at high speed, up to 10's GHz [3]. High-speed capability can be achieved by designing electrodes for high-speed operation (radio frequency electrodes) and 100's GHz operating frequency is expected to be achieved with thin film lithium niobate technology [4].

The universal processors the referee discusses are 'digital' or 'discrete' networks that realize a unitary transformation, but not Hamiltonian continuous-time evolution, and all are based on the thermo-optic effect. The discrete approach is fundamentally different than the continuously coupled one, the latter presenting alternative opportunities for simulating physics or manipulating states of light [5].

Waveguide arrays can also be cascaded to obtain universality. This was shown in [6]. Our device is the basic building block of a universal circuit based on continuously coupled circuits, in contrast to the beam splitter based approaches (a.k.a. Reck/Clements' schemes). We highlighted in our paper a number of advantages related to the former: 1) significantly reduces the number of bends, which reduces loss and footprint [7, 8], 2) Direct simulation and control of Hamiltonian evolution, 3) Significantly reduced error rate as presented in [6]. Additionally, our device is ideal for studying complex time evolution dynamics of arbitrary Hamiltonians, including continuous simulation of time, which cannot be performed via fabricating separated static devices. On the other hand, current-available platforms lack compatibility with high-speed reconfiguration, bend loss-free operation, and low error rates.

Regarding fidelity, this is the first prototype of a device presenting this new capability. This work was focused on showcasing this technology and its potential for performing high-fidelity controllable simulations and information processing. We expect future devices to achieve higher fidelity.

We have changed the following text at the end of the 'Discussion' section:

The wave equation in the paraxial and scalar approximations is similar to the Schrodinger equation where the refractive index profile plays the role of a potential. This is why waveguide array structures have been used to simulate solid-state quantum phenomena with laser light [9]. Furthermore, a classical (laser light) characterization of the Hamiltonian completely predicts the device performance with quantum states. Therefore, with the ability to independently and accurately control the propagation and coupling parameters of a waveguide array, this new structure can be used for a variety of applications, ranging from simulating complex physics systems [10–13] to processing quantum information [6–8, 14–16] as well as continuous simulation of time for complex time evolution dynamics of arbitrary Hamiltonians.

We have changed the following text and references in the 'Introduction' section:

optical transformation [15] and could implement 1 and 2-qubit gates [14, 17] with the potential of implementing universal unitaries [6]

We have added reference [18] in the following text in the 'Introduction' section:

Waveguide arrays [19] are a powerful platform for the optical simulation of condensed matter physics effects ranging from Bloch oscillations [20], to enhanced coherent transport via controllable decoherence [21], adiabatic passage [22, 23], Anderson localization [24], and many more [18].

We have added references [25, 26] in the following text in the 'Introduction' section:

These devices actualize either single-particle unitaries through cascaded, programmable Mach-Zehnder interferometers (MZI) [25–27], or a single-particle Hamiltonian that is determined by the detailed configuration of the waveguides.

We have added the following text and references [28] in the 'Discussion' section:

Achieving scalability is the prime focus in advanced integrated photonic systems. Despite the capability of femtosecond-laser-written waveguides [28] to implement arbitrary or time-dependent Hamiltonians, they are limited as non-universal static devices. Furthermore, although larger circuits incorporating MZIs as fundamental reconfigurable units have been extensively demonstrated [25, 26], the scalability of MZIs is impeded by their sensitivity to fabrication errors and their susceptibility to significant bending loss [29, 30].

Comment: *The authors also mention the suppression of cross-talks among the potentials of their device. This is expected to guarantee high fidelities. However, I observe that the measured fidelities in reaching the theoretical distributions at the output are not exceptionally high, especially if compared to the mentioned works by Taballione et al. or by Harris et al. On this aspect one can also note the non-negligible discrepancy in the measured/simulated participation ratios (PR) in Figure 3d. Since the PR is a collective figure of merit that should be quite robust to imperfections, it may be an indication that control on the waveguide parameters is only obtained with some approximation.*

Response:

As we mentioned in the previous response, this work showcases the first prototype of a device presenting this new capability. It is anticipated that forthcoming iterations of this device will attain greater fidelity.

We changed the sentence below in the 'Discussion' section to address the comment on fidelity:

Additionally, using more advanced fabrication methods, particularly for the electrodes, will also help improve performance and scalability. In particular, the thin film lithium niobate platform may allow reducing the footprint—enabling the fabrication of cascaded RWAs—while keeping the advantage of electro-optic performance including high-speed operations [4, 31–33].

Comment: *In the end, I think it cannot be concluded that this electro-optic device provides outstanding applicative advantages with respect to the other photonic processors reported in the literature. I would not recommend publication of this manuscript in Nature Communications, but rather I think this work would deserve to be showcased in a more specialized journal, where the photonic-technology aspects would likely find a solid appreciation.*

Response: We appreciate the reviewer's thoughtful assessment and remarks of our work, although we disagree with their assessment regarding suitability for Nature Communications.

We believe we fully and satisfactorily addressed the referee's comments above, and that the improved manuscript meets the scope of and matches the body of work published in Nature Communications.

Our study presents the first prototype of a programmable waveguide array in which the Hamiltonian terms can be continuously electro-optically tuned to implement various Hamiltonian continuous-time evolutions on a single device. We believe this makes it a significant advancement to the existing knowledge in this area. Specifically, this platform provides substantial progress and enables new opportunities including:

- implementation of universal unitaries with improved fabrication error tolerance and significantly reduced bending loss [6–8, 15, 16]
- high-speed multimode routing and mode division multiplexing [17, 34–36],
- Hamiltonian continuous-time evolutions implementation with precisely controlled Hamiltonian parameters, as an alternative approach to the discrete implementation [25],
- simulations of higher-dimensional systems such as in recent proposal [12, 13]
- possibilities of demonstrating more complex experiments [10–13], as well as continuous simulation of time for arbitrary Hamiltonians time evolution dynamics.

Comment: *In addition, I would strongly suggest the authors to address the following points, which in my opinion are essential for a fair assesment of the quality of the reported results:*

1) *the operating wavelength should be clearly stated;*

Response:

In the paper, we stated that 'An 808 nm polarized laser' was used in the caption of Supplementary Fig 1 and in the section 'Measurement and control setup' in the Supplementary Methods.

We added the following sentence in the 'Details of the device' section in the Supplementary Methods:

and we used an 808 nm polarized laser

Comment: *2) characterization of the device insertion loss should be reported and discussed; if possible, an estimate of the different loss contributions should also be given (e.g. input/output coupling loss due to mode matching, propagation loss per unit length of the waveguide...)*

Response: We thank the referee for the suggestions. We improved the following sentence in the 'Measurement and control setup' section in the Supplementary Methods:

Polarization-maintaining fiber (PMF) arrays with 127 μm pitch were used for butt-coupling to the chip at the input and output sides with a coupling loss of 4.9 dB6 per facet (68%) due to fiber and waveguide mode mismatching.

Comment: *3) the authors should discuss with some detail how they have calibrated precisely the action of each electrode, namely how they have estimated the coefficients $\Delta\beta_n$, ΔC_1 and ΔC_2 in Eq. (3) and (4).*

Response: To reconstruct the Hamiltonian dependence on the control voltages, we used the data from the Aubry-André and SSH experiment to fit the models described in Equations 1-6. We assumed β_0 , $\Delta\beta_n$, C_0 , ΔC_1 and ΔC_2 to be consistent across the array, as discussed in the 'Waveguide array overview and modeling' section in the main text.

The fitting fidelity was limited likely by small differences between the electrodes, which can be attributed to either fabrication or control imperfections. We reported the fitted values in the 'Details of the Device' section in the Supplementary Methods.

As we stated in the 'Discussion' section in the main text

Firstly, fabrication errors in the waveguides, such as the waveguide width and separation, can cause β_0 and C_0 to be not identical across the array. Secondly, the dimensions, position, and quality of the fabricated electrodes can vary, making $\Delta\beta_n$ and $\Delta C_{n,n+1}$ site-dependent.

Comment: *4) the uniformity of the static and reconfigurable waveguide parameters along the propagation coordinate should be discussed or analyzed. For instance, the authors might implement a nominally uniform*

Hamiltonian, measuring the amount of "noise" in the output distribution and fitting it back to simulated models. Other more appropriate ways could anyway be devised and adopted.

Response: We have addressed this issue in the previous comment regarding fabrication technology and in the below sentence in the 'Discussion' section in the main text:

Many experimental factors also affect the model fidelity. Firstly, fabrication errors in the waveguides, such as the waveguide width and separation, can cause β_0 and C_0 to be not identical across the array. Secondly, the dimensions, position, and quality of the fabricated electrodes can vary, making $\Delta\beta_n$ and $\Delta C_{n,n+1}$ site-dependent.

We improved the sentence in the 'Simulation of the voltage-controlled output distribution for the Aubry-André and SSH Hamiltonians' section in the Supplementary Methods:

The fidelity for the Aubry-André and SSH models is 0.949 ± 0.009 and 0.904 ± 0.001 respectively.

and have added the following sentence in the same section:

The static Hamiltonians fitting fidelity is 0.878 ± 0.001 with parameters fitted from Aubry-André and SSH model measurements.

We improved the following sentence in the main text accordingly to the 'Aubry-André model' section:

The fidelity with the theoretical model is 0.949 ± 0.009 (see the simulation in Supplementary Methods).

and to 'Su-Schrieffer-Heeger model' section:

The fidelity with the theoretical model is 0.904 ± 0.001 (see the simulation in Supplementary Methods).

Comment: *5) the time-response of the device should be characterized experimentally.*

Response: We thank the reviewer for the suggestion. The time-response of the device was indeed characterized, and it is related to the design of the waveguides and electrodes.

In this work, our goal was to mitigate the drift in the optical response caused by electric charge diffusion in the Lithium Niobate (DC drift [37]). In the 'Measurement and control setup' section in Supplementary Methods, we discussed that we achieved this using modulated square pulses with a 0.6 Hz frequency. Higher operating frequencies are expected to be achieved as discussed in the previous response.

To address the referee's suggestion, we added Supplementary Fig 2 showing the time response of our device for a single pulses electrode, and we improved the following paragraph:

Lithium niobate suffers from electric charges accumulating in the SiO_2 buffer layer under the control electrodes, which results in a drift of the optical output when a voltage is applied. To mitigate the output optical drift, 1.66 s non-biased square pulses were applied in order to achieve unbiased control, i.e. each target voltage is followed by a pulse of the same magnitude and opposite sign (see Supplementary Fig 2 (a)).

and we added a sentence in the same section:

In Supplementary Fig 2 (b), we report the chip response of the pulsing scheme used to reset the chip to the initial state (where the Hamiltonian is not affected by the voltage). The data for each measurement is taken from the average of the output power in the first half of the non-biased square pulse after the rising time (~ 0.22 s), which is limited by the built-in low-pass filter of the multi-channel power meter. We characterized the rise time as less than 0.2 ms based on optical response with fast photodiodes, and optical outputs keep drifting due to electric charging after the rise time.

Referee 3

Comment: *In this work, Yang et al., demonstrate a reconfigurable waveguide array fabricated in an annealed proton exchange lithium-niobate platform. The authors exploit the reconfigurability of their system to implement different input/output transformations corresponding to different Hamiltonians. Specifically, they consider the Su-Schrieffer-Heeger (SSH) model, the Aubrey-Andre model, and Anderson localization. The linear characterization of the transmission function of the structure shows a reliable and effective reconfigurability of their structure. Fidelities between the expected and measured transmission function are above 90% for all the examined configurations. The main claim of the manuscript is that this platform can find applications for quantum information processing in high-dimensional systems.*

These experimental results are solid and convincing to demonstrate that their platform is flexible and reconfigurable in a predictive way.

Response: We thank the reviewer for their time and effort in reviewing our manuscript and providing feedback.

Comment: *However, the claim that this platform will have an impact on quantum information processing is not demonstrated. First, all the experimental results presented in the manuscript are obtained with classical light.*

Response: We thank the referee for pointing this out.

The wave equation in the paraxial and scalar approximations is similar to the Schrodinger equation where the refractive index profile plays the role of a potential. This is why waveguide array structures have been used to simulate solid-state quantum phenomena with laser light [Longhi et al, PHYSICAL REVIEW B 74, 155116 (2006)]. Furthermore, a classical (laser light) characterization of the Hamiltonian completely predicts the device performance with quantum states.

However, our device can find applications and has a potential impact far beyond the field of quantum science and technology. We have therefore reduced the focus on quantum applications to make it more general.

We corrected the abstract:

We present a programmable waveguide array in which the Hamiltonian terms can be electro-optically tuned to implement various Hamiltonian continuous-time evolutions on a single device.

We changed the following text in the 'Introduction' section:

Achieving such control is a key step toward versatile photonic processors, routers, and simulators [6–8, 10–17, 34, 36].

and

We demonstrate precise control over independent Hamiltonian terms to realize continuous-time evolutions for several thousands of Hamiltonian.

and we corrected the following sentence in the 'Waveguide array overview and modeling' section:

To implement the control, the conventional thermo-optic effect employed in most integrated photonic circuits reported so far, cannot be used since with such a small gap between the electrodes, thermal cross-talk would be unavoidable [2]. In this device, the high electro-optic coefficient of lithium niobate allows ultra-high modulation speed with almost no cross-talk and power dissipation [1]. This assumption is based on the exceptional confinement of the electric field within the material due to the shielding effect from neighboring electrodes, thereby preventing cross-talk with other waveguides in the array.

We changed the opening sentence in the 'Discussion' section:

Reconfigurable waveguide arrays are a powerful tool for optical signal routing, implementing time-evolution Hamiltonians to simulate quantum materials and phenomena, and can be cascaded to realize arbitrary unitaries.

and the last paragraph in the ‘Discussion’ section:

The wave equation in the paraxial and scalar approximations is similar to the Schrodinger equation where the refractive index profile plays the role of a potential. This is why waveguide array structures have been used to simulate solid-state quantum phenomena with laser light [9]. Furthermore, a classical (laser light) characterization of the Hamiltonian completely predicts the device performance with quantum states. Therefore, with the ability to independently and accurately control the propagation and coupling parameters of a waveguide array, this new structure can be used for a variety of applications, ranging from simulating complex physics systems [10–13] to processing quantum information [6–8, 14–16] as well as continuous simulation of time for complex time evolution dynamics of arbitrary Hamiltonians.

Comment: *Second, it is not clear what are the limits of the coupled-waveguide approach in terms of the Hamiltonian can be realized. It seems as if the authors chose SSH and Aubrey-Andre models because they are the easiest to be studied in coupled waveguide arrays, however, their relevance for quantum information processing is not clear.*

Response: This work is focused on demonstrating precise control over independent Hamiltonian terms of an RWA device, and the Hamiltonians used here were selected because of their wide application in the literature by the use of static devices.

Our platform, when cascaded in series, could implement arbitrary unitaries [6], and integrated photonic circuits process both quantum and classical information. Recent theoretical proposals [6–8, 14, 15], have suggested several quantum applications of this technology.

Comment: *Third, the scalability of the approach is not clear. This ultimately depends on the geometrical configuration (coupled waveguide array), the refractive index contrast, and the maximum length of the system. How many waveguides one can realistically implement? Other details regarding possible issues, such as DC drift in LiNbO₃, are not discussed. These are crucial aspects given the ambitious claims of the authors.*

Response: Several theoretical works have discussed the scalability of using WAs or RWAs as a platform [6, 15, 16] and there is no fundamental problem with building RWA with hundreds of spatial modes and cascading them to implement larger scale circuits.

In the ‘discussion’ section, we suggested that *the lithium niobate in insulator platform may allow reducing the footprint—enabling the fabrication of cascaded RWAs—while keeping the advantage of electro-optic performance including high-speed operations [4, 31–33].*

We have addressed DC drift in the ‘Measurement and control setup’ section in Supplementary Methods. To improve this discussion, we have added a new plot and the following text:

To avoid this problem, 1.66 s non-biased square pulses were applied in order to achieve unbiased control, i.e. each target voltage is followed by a pulse of the same magnitude and opposite sign (see Supplementary Fig 2 (a))

and

In Supplementary Fig 2 (b), we report the chip response of the pulsing scheme used to reset the chip to the initial state (where the Hamiltonian is not affected by the voltage). The data for each measurement is taken from the average of the output power in the first half of the non-biased square pulse after the rising time (~ 0.22 s) caused by the built-in low-pass filter of the multi-channel power meter. We characterized the rise time as less than 0.2 ms based on optical response with fast photodiodes, and optical outputs keep drifting due to electric charging after the rise time.

Comment: *The manuscript is sometimes a bit misleading. Indeed, in the introduction, the authors cite several papers presenting platforms in which quantum experiments have been carried out. The conclusions also point in the direction of quantum information processing. However, all the experimental results are obtained using classical light, to which the authors often refer as “input light”, without specifying or stressing that it is classical in nature. While such a linear characterization is important, it is not sufficient to establish what the performances with nonclassical light would be.*

Response: We note that complete knowledge of linear optical circuit transformation can be obtained with classical light [Rahimi Keshari et al, 2013/6/3 Optics express 21, 13450-13458]. This suffices to predict the device performance with quantum states.

We believe the changes we applied in the introduction make the scope of the paper broader than the original focus on quantum mechanical systems.

Comment: *The authors discuss advantages such as the possibility of integrating single-photon sources and detectors on chips. However, these are not demonstrated in the present system.*

Response:

This sentence in the ‘Discussion’ section serves as an outlook into opportunities for expanding the capability of this technology, which is common in reports on experimental demonstrations.

We have expanded the outlook with more references [38, 39] and addressed the issue in the following text in the ‘Discussion’ section:

Moreover, it offers enhanced nonlinearity [40], the possibility of integrating single-photon sources [38, 41] and detectors [42] on chips, cryogenic compatible operations, increasing the scalability and reducing the coupling losses.

Comment: *Given the numerous experimental demonstrations of quantum light manipulation in complex photonic integrated circuits in various material platforms, I am afraid the results reported here are not significant enough to grant publication in Nature Communication. I believe that the manuscript is more suitable for other kinds of journals, e.g., Applied Physics Letter or Optics Express.*

Response: We appreciate the reviewer’s thoughtful assessment of our work and their suggestions.

We believe we fully and satisfactorily addressed the referee’s comments above, and that the improved manuscript meets the scope of and matches the body of work published in Nature Communications.

Our study presents a programmable waveguide array in which the Hamiltonian terms can be electro-optically continuously tuned to implement various Hamiltonian continuous-time evolutions on a single device. We believe this makes it a significant advancement to the existing knowledge in this area. Specifically, this platform provides substantial progress and enables new opportunities including:

- implementation of universal unitaries with improved fabrication error tolerance and significantly reduced bending loss [6–8, 15, 16],
- high-speed multimode routing and mode division multiplexing [17, 34–36],
- Hamiltonian continuous-time evolutions implementation with precisely controlled Hamiltonian parameters, as an alternative approach to the discrete implementation [25],
- simulations of higher-dimensional systems such as in recent proposal [12, 13],
- possibilities of demonstrating more complex experiments [10–13], as well as continuous simulation of time for arbitrary Hamiltonians time evolution dynamics.

-
- [1] Alessandro Prencipe and Katia Gallo. Electro- and thermo-optics response of x-cut thin film linbo3 waveguides. *IEEE Journal of Quantum Electronics*, 59(3):1–8, 2023.
- [2] Francesco Ceccarelli, Simone Atzeni, Ciro Pentangelo, Francesco Pellegatta, Andrea Crespi, and Roberto Osellame. Low power reconfigurability and reduced crosstalk in integrated photonic circuits fabricated by femtosecond laser micromachining. *Laser & Photonics Reviews*, 14(10):2000024, 2020.
- [3] Yifan Qi and Yang Li. Integrated lithium niobate photonics. *Nanophotonics*, 9(6):1287–1320, 2020.
- [4] Cheng Wang, Mian Zhang, Xi Chen, Maxime Bertrand, Amirhassan Shams-Ansari, Sethumadhavan Chandrasekhar, Peter Winzer, and Marko Lončar. Integrated lithium niobate electro-optic modulators operating at CMOS-compatible voltages. *Nature*, 562(7725):101–104, October 2018. Number: 7725 Publisher: Nature Publishing Group.
- [5] Tomoki Ozawa, Hannah M. Price, Alberto Amo, Nathan Goldman, Mohammad Hafezi, Ling Lu, Mikael C. Rechtsman, David Schuster, Jonathan Simon, Oded Zilberberg, and Iacopo Carusotto. Topological photonics. *Reviews of Modern Physics*, 91(1):015006, 2019. Publisher: American Physical Society.
- [6] M. Yu. Saygin, I. V. Kondratyev, I. V. Dyakonov, S. A. Mironov, S. S. Straupe, and S. P. Kulik. Robust architecture for programmable universal unitaries. *Phys. Rev. Lett.*, 124:010501, Jan 2020.
- [7] J. Petrovic and J. J. P. Veerman. A new method for multi-bit and qudit transfer based on commensurate waveguide arrays. *Annals of Physics*, 392:128–141, May 2018.
- [8] Jovana Petrovic, Jelena Krsic, Peter J. J. Veerman, and Aleksandra Maluckov. A new concept for design of photonic integrated circuits with the ultimate density and low loss, 2021.
- [9] S. Longhi, M. Lobino, M. Marangoni, R. Ramponi, P. Laporta, E. Cianci, and V. Foglietti. Semiclassical motion of a multiband bloch particle in a time-dependent field: Optical visualization. *Phys. Rev. B*, 74:155116, Oct 2006.
- [10] Miguel A. Bandres, Oded Zilberberg, and Andrey Sukhorukov. Special topic on synthetic gauge field photonics. *APL Photonics*, 7(5):050401, 05 2022.
- [11] Javier del Pino and Oded Zilberberg. Dynamical gauge fields with bosonic codes. *Phys. Rev. Lett.*, 130:171901, Apr 2023.
- [12] Stefano Longhi. Photonic simulation of giant atom decay. *Optics Letters*, 45(11):3017–3020, June 2020. Publisher: Optica Publishing Group.
- [13] Lukas J. Maczewsky, Kai Wang, Alexander A. Dovgii, Andrey E. Miroshnichenko, Alexander Moroz, Max Ehrhardt, Matthias Heinrich, Demetrios N. Christodoulides, Alexander Szameit, and Andrey A. Sukhorukov. Synthesizing multi-dimensional excitation dynamics and localization transition in one-dimensional lattices. *Nature Photonics*, 14(2):76–81, February 2020.
- [14] Yoav Lahini, Gregory R. Steinbrecher, Adam D. Bookatz, and Dirk Englund. Quantum logic using correlated one-dimensional quantum walks. *npj Quantum Information*, 4(1):1–7, 2018.
- [15] N. N. Skryabin, N. N. Skryabin, I. V. Dyakonov, M. Yu Saygin, and S. P. Kulik. Waveguide-lattice-based architecture for multichannel optical transformations. *Optics Express*, 29(16):26058–26067, 2021. Publisher: Optica Publishing Group.
- [16] Ryota Tanomura, Rui Tang, Toshikazu Umezaki, Go Soma, Takuo Tanemura, and Yoshiaki Nakano. Scalable and Robust Photonic Integrated Unitary Converter Based on Multiplane Light Conversion. *Physical Review Applied*, 17(2):024071, February 2022. Publisher: American Physical Society.
- [17] Enrico Compagno, Leonardo Banchi, and Sougato Bose. Toolbox for linear optics in a one-dimensional lattice via minimal control. *Physical Review A*, 92(2):022701, 2015. Publisher: American Physical Society.
- [18] S. Longhi. Quantum-optical analogies using photonic structures. *Laser & Photonics Reviews*, 3(3):243–261, 2009.
- [19] Demetrios N. Christodoulides, Falk Lederer, and Yaron Silberberg. Discretizing light behaviour in linear and nonlinear waveguide lattices. *Nature*, 424(6950):817–823, 2003.
- [20] R. Morandotti, U. Peschel, J. S. Aitchison, H. S. Eisenberg, and Y. Silberberg. Experimental observation of linear and nonlinear optical bloch oscillations. *Phys. Rev. Lett.*, 83:4756–4759, Dec 1999.
- [21] Devon N. Biggerstaff, René Heilmann, Aidan A. Zecevik, Markus Gräfe, Matthew A. Broome, Alessandro Fedrizzi, Stefan Nolte, Alexander Szameit, Andrew G. White, and Ivan Kassal. Enhancing coherent transport in a photonic network using controllable decoherence. *Nature Communications*, 7(1), apr 2016.
- [22] Emmanuel Paspalakis. Adiabatic three-waveguide directional coupler. *Optics Communications*, 258(1):30–34, 2006.
- [23] Y. Lahini, F. Pozzi, M. Sorel, R. Morandotti, D. N. Christodoulides, and Y. Silberberg. Effect of nonlinearity on adiabatic evolution of light. *Phys. Rev. Lett.*, 101:193901, Nov 2008.
- [24] Yoav Lahini, Assaf Avidan, Francesca Pozzi, Marc Sorel, Roberto Morandotti, Demetrios N. Christodoulides, and Yaron Silberberg. Anderson localization and nonlinearity in one-dimensional disordered photonic lattices. *Physical Review Letters*, 100(1):013906, 2008. Publisher: American Physical Society.
- [25] Nicholas C. Harris, Gregory R. Steinbrecher, Mihika Prabhu, Yoav Lahini, Jacob Mower, Darius Bunandar, Changchen Chen, Franco N. C. Wong, Tom Baehr-Jones, Michael Hochberg, Seth Lloyd, and Dirk Englund.

- Quantum transport simulations in a programmable nanophotonic processor. *Nature Photonics*, 11(7):447–452, 2017.
- [26] Caterina Taballione, Malaquias Correa Anguita, Michiel de Goede, Pim Venderbosch, Ben Kassenberg, Henk Sniijders, Narasimhan Kannan, Ward L. Vleeshouwers, Devin Smith, Jörn P. Epping, Reinier van der Meer, Pepijn W. H. Pinkse, Hans van den Vlekkert, and Jelmer J. Renema. 20-mode universal quantum photonic processor, 2023.
- [27] Jacques Carolan, Christopher Harrold, Chris Sparrow, Enrique Martín-López, Nicholas J. Russell, Joshua W. Silverstone, Peter J. Shadbolt, Nobuyuki Matsuda, Manabu Oguma, Mikitaka Itoh, Graham D. Marshall, Mark G. Thompson, Jonathan C. F. Matthews, Toshikazu Hashimoto, Jeremy L. O'Brien, and Anthony Laing. Universal linear optics. *Science*, 349(6249):711–716, August 2015.
- [28] Alexander Szameit and Stefan Nolte. Discrete optics in femtosecond-laser-written photonic structures. *Journal of Physics B: Atomic, Molecular and Optical Physics*, 43(16):163001, jul 2010.
- [29] David A. B. Miller. Perfect optics with imperfect components. *Optica*, 2(8):747, aug 2015.
- [30] Roel Burgwal, William R. Clements, Devin H. Smith, James C. Gates, W. Steven Kolthammer, Jelmer J. Renema, and Ian A. Walmsley. Using an imperfect photonic network to implement random unitaries. *Opt. Express*, 25(23):28236–28245, Nov 2017.
- [31] Mian Zhang, Cheng Wang, Rebecca Cheng, Amirhassan Shams-Ansari, and Marko Lončar. Monolithic ultra-high-Q lithium niobate microring resonator. *Optica*, 4(12):1536 – 1537, 00 2017.
- [32] Inna Krasnokutska, Jean-Luc J. Tambasco, Xijun Li, and Alberto Peruzzo. Ultra-low loss photonic circuits in lithium niobate on insulator. *Optics Express*, 26(2):897–904, January 2018. Publisher: Optical Society of America.
- [33] Inna Krasnokutska, Jean-Luc J. Tambasco, and Alberto Peruzzo. Tunable large free spectral range microring resonators in lithium niobate on insulator. *Scientific Reports*, 9(1):11086, 06 2018.
- [34] Alastair Kay. Perfect state transfer: Beyond nearest-neighbor couplings. *Phys. Rev. A*, 73:032306, Mar 2006.
- [35] Philippe Velha, Isabella Cerutti, and Nicola Andriolli. Crosstalk and ber performance of closely-spaced silicon-on-insulator waveguide arrays. *Optics Communications*, 437:214–218, 2019.
- [36] N. N. Skryabin, I. V. Dyakonov, M. Yu. Saygin, and S. P. Kulik. Waveguide-lattice-based architecture for multichannel optical transformations. *Opt. Express*, 29(16):26058–26067, Aug 2021.
- [37] Syoji Yamada and Makoto Minakata. DC Drift Phenomena in LiNbO3 Optical Waveguide Devices. *Japanese Journal of Applied Physics*, 20(4):733, April 1981. Publisher: IOP Publishing.
- [38] Daniel White, Artur Branny, Robert J. Chapman, Raphaël Picard, Mauro Brotons-Gisbert, Andreas Boes, Alberto Peruzzo, Cristian Bonato, and Brian D. Gerardot. Atomically-thin quantum dots integrated with lithium niobate photonic chips. *Opt. Mater. Express*, 9(2):441–448, Feb 2019.
- [39] Patrik I. Sund, Emma Lomonte, Stefano Paesani, Ying Wang, Jacques Carolan, Nikolai Bart, Andreas D. Wieck, Arne Ludwig, Leonardo Midolo, Wolfram H. P. Pernice, Peter Lodahl, and Francesco Lenzini. High-speed thin-film lithium niobate quantum processor driven by a solid-state quantum emitter. *Science Advances*, 9(19):eadg7268, 2023.
- [40] Marc Jankowski, Carsten Langrock, Boris Desiatov, Alireza Marandi, Cheng Wang, Mian Zhang, Christopher R. Phillips, Marko Lončar, and M. M. Fejer. Ultrabroadband nonlinear optics in nanophotonic periodically poled lithium niobate waveguides. *Optica*, 7(1):40–46, Jan 2020.
- [41] Shahriar Aghaeimebodi, Boris Desiatov, Je-Hyung Kim, Chang-Min Lee, Mustafa Atabey Buyukkaya, Aziz Karasahin, Christopher J. K. Richardson, Richard P. Leavitt, Marko Lončar, and Edo Waks. Integration of quantum dots with lithium niobate photonics. *Applied Physics Letters*, 113(22):221102, 11 2018.
- [42] Emma Lomonte, Martin A. Wolff, Fabian Beutel, Simone Ferrari, Carsten Schuck, Wolfram H. P. Pernice, and Francesco Lenzini. Single-photon detection and cryogenic reconfigurability in lithium niobate nanophotonic circuits. *Nature Communications*, 12(1):6847, 03 2021.

REVIEWER COMMENTS

Reviewer #1 (Remarks to the Author):

I thank the authors for the response to my previous comments and observations. I find that the authors have satisfactorily responded to my previous remarks.

Given my previous assessment of the manuscript, thus considering the relevance of implementing interferometers with electro-optic technology for single-waveguide reconfiguration capabilities, I can thus support its publication in *Nature Communications*.

Reviewer #2 (Remarks to the Author):

I have read with interest the Authors' reply, and I have seen the modifications brought to the manuscript in response to my comments, as well as to the comments of the other reviewers.

I am still not fully convinced that the device concept reported here enables, at present, game-changing features with respect to other reconfigurable photonic architectures based e.g. on the well-known Clements' decomposition.

However, the Authors' reply together with the revised manuscript make me appreciate more clearly the substantial technological advancements that the Authors have achieved. As a matter of fact, the reconfiguration of a waveguide array had never been demonstrated with this degree of detail and within the coupling region. While the present realization may not bring yet big advantages in quantum information applications (the reported experiments are indeed performed with classical light), these results are indeed significant and impactful in the wider field of photonics.

I can now support the publication of this manuscript in *Nature Communications*, provided that the following two points are fixed.

- 1) The type of light source used for the experiments, with its wavelength, should be mentioned clearly and explicitly in the Main Text.
- 2) The Authors should declare also the other loss contributions of the waveguide device, or at least the full insertion loss, not just the coupling losses at the input/output facets.

Reviewer #3 (Remarks to the Author):

I am writing to submit my second-round review of the manuscript titled "Programmable High-Dimensional Hamiltonian in a Photonic Waveguide Array." While I appreciate the authors' efforts in revising the manuscript in response to my concerns and those of the other two reviewers, I do not believe that they have adequately addressed my comments in this revised version. For this reason, I cannot support the publication of the manuscript in *Nature Communication*, I still believe that this work would be more suitable for Applied Physics Letter or Optics Express.

My main concern, also shared by Reviewer #2, is whether this work makes a sufficient case to demonstrate that this platform is capable to access new and relevant experimental results in the field of photonic computing, whether classical or quantum. As I pointed out in my previous report, the authors are dealing with a technological platform (proton-exchange LiNbO waveguides) that is relatively established, and they demonstrate a system comprising 11 tunable waveguides, which is not sufficient to access new regimes or stimulate interesting new phenomena. The key problem is understanding how far we can go in this direction. In this respect, this second version of the

manuscript does not provide any new element. In my previous report, I asked to provide more details, for example, a reasonable number of waveguides that one could implement, but this question remained unanswered.

I find it somewhat curious that in the Discussion section, the authors suggest that the scalability of their approach may be achieved in the thin-film LiNbO₃ platform, where a more compact design could be implemented. This is strange, not only because this is not the approach investigated here, but because on page 2 the same authors write: "To avoid voltage breakdown through air and glass, the electrodes must be patterned at sufficient separation. This is possible because the large cross-section of the mode in a proton-exchange waveguide leads to substantial coupling between waveguides more than 10 μm apart," which seems to indicate an advantage of well-separated waveguides. Furthermore, implementing this idea in a different platform would certainly require addressing new experimental issues, (e.g. electrode configuration), which are not mentioned at all.

Another concern raised in my previous report was related to universality. In their reply, the authors state that "our platform, when cascaded in series, could implement arbitrary unitaries [6]" While this statement holds true in theory, the practical implications remain unclear. How many distinct sections are necessary? What length must each section have? Is the approach feasible in terms of minimizing losses? Unfortunately, the authors' response does not offer sufficient information to establish the scalability and universality of their approach. Thus, the impact of their experimental result, which is one of the main issues, remains unclear.

Previously, I highlighted that the experimental results presented in this work are purely classical in nature. I appreciate that the authors made this clear in this second version of the manuscript by removing/modifying some of their statements. However, their assertion, "Furthermore, a classical (laser light) characterization of the Hamiltonian completely predicts the device performance with quantum states," can be misleading. While it is indeed possible to determine the device's transfer function using classical light, it is important to recognize that the device performance is influenced also by the quantum state and the specific experimental conditions. For instance, the spectral properties of the photons will play a crucial role in determining the visibility of quantum interference, depending upon the waveguide dispersion.

It should be noted that the experiment described in the paper is conducted at a single wavelength (808 nm), a detail which regrettably goes unmentioned in the main text. Furthermore, the fidelities measured in this work may not be meaningful in a quantum experiment. Indeed, the device is burdened with significant insertion losses, which could render quantum experiments impractical. The prospect of low detection rates, for example, could necessitate excessively lengthy experiments, potentially impacting the device's stability in ways not thoroughly investigated in this study. Therefore, if the intention is to assert the device's viability for quantum experiments, it becomes imperative to conduct such experiments. This is because a host of complexities exist beyond a mere linear characterization using continuous-wave laser light.

Referee 1

Comment: *I thank the authors for the response to my previous comments and observations. I find that the authors have satisfactorily responded to my previous remarks. Given my previous assessment of the manuscript, thus considering the relevance of implementing interferometers with electro-optic technology for single-waveguide reconfiguration capabilities, I can thus support its publication in Nature Communications.*

Response: We thank the reviewer for their time and effort in reviewing our response letter and revised manuscript, and for supporting publication in *Nature Communications*.

Referee 2

Comment: *I have read with interest the Authors' reply, and I have seen the modifications brought to the manuscript in response to my comments, as well as to the comments of the other reviewers. I am still not fully convinced that the device concept reported here enables, at present, game-changing features with respect to other reconfigurable photonic architectures based e.g. on the well-known Clements' decomposition. However, the Authors' reply together with the revised manuscript make me appreciate more clearly the substantial technological advancements that the Authors have achieved. As a matter of fact, the reconfiguration of a waveguide array had never been demonstrated with this degree of detail and within the coupling region. While the present realization may not bring yet big advantages in quantum information applications (the reported experiments are indeed performed with classical light), these results are indeed significant and impactful in the wider field of photonics.*

I can now support the publication of this manuscript in Nature Communications, provided that the following two points are fixed.

Response: We thank the referee for their time and effort in reviewing our response letter and revised manuscript, and for supporting publication in *Nature Communications*. We have addressed their comments below.

Comment: *1) The type of light source used for the experiments, with its wavelength, should be mentioned clearly and explicitly in the Main Text.*

Response: We have added the following text at the end of the 'Waveguide array overview and modeling' section:

The light source used in this work is a fiber-coupled polarized 808 nm laser diode.

Comment: *2) The Authors should declare also the other loss contributions of the waveguide device, or at least the full insertion loss, not just the coupling losses at the input/output facets.*

Response: We thank the referee for the suggestions. We have improved the following text and added the detail of full insertion loss in the 'Measurement and control setup' section in the Supplementary Methods:

A polarized 808 nm laser and multi-channel fiber-coupled high-speed optical power meter were used for the output measurements. Polarization-maintaining fiber (PMF) arrays with 127 μm pitch were used for butt-coupling to the chip at the input and output sides with a coupling loss of 4.9 dB per facet (68%), caused by fiber-waveguide mode mismatch. A total power loss, including the fiber-to-chip coupling and propagation loss, of 9.8 dB (90%) was measured. This is sufficient for two-photon quantum experiments, as demonstrated in the first two-photon quantum walk in a static waveguide array [1]. To enhance the total transmission of the device, for example for experiments using more than two photons, the insertion loss can be reduced by various technologies, ranging from on-chip components to engineered fibers, such as high-index fibers that reduce the mode field diameter of the fiber [2–6].

Referee 3

Comment: *I am writing to submit my second-round review of the manuscript titled "Programmable High-Dimensional Hamiltonian in a Photonic Waveguide Array." While I appreciate the authors' efforts in revising the manuscript in response to my concerns and those of the other two reviewers, I do not believe that they have adequately addressed my comments in this revised version. For this reason, I cannot support the publication of the manuscript in Nature Communication, I still believe that this work would be more suitable for Applied Physics Letter or Optics Express.*

My main concern, also shared by Reviewer #2, is whether this work makes a sufficient case to demonstrate that this platform is capable to access new and relevant experimental results in the field of photonic computing, whether classical or quantum.

Response: We appreciate the reviewer's thoughtful assessment of our work and their suggestions.

We politely disagree with the reviewer's opinion of our work. After our manuscript revision, Reviewers #1 and #2 now fully support publication in *Nature Communications*.

Reviewer #1 stated:

Given my previous assessment of the manuscript, thus considering the relevance of implementing interferometers with electro-optic technology for single-waveguide reconfiguration capabilities, I can thus support its publication in Nature Communications.

Reviewer #2 stated:

...the reconfiguration of a waveguide array had never been demonstrated with this degree of detail and within the coupling region. While the present realization may not bring yet big advantages in quantum information applications (the reported experiments are indeed performed with classical light), these results are indeed significant and impactful in the wider field of photonics. I can now support the publication of this manuscript in Nature Communications, ...

To reiterate, our study presents the first programmable waveguide array in which the Hamiltonian terms can be precisely controlled electro-optically to implement various Hamiltonian continuous-time evolutions on a single device. We believe this makes it a significant advancement to the existing literature in this area.

Comment: *As I pointed out in my previous report, the authors are dealing with a technological platform (proton-exchange LiNbO waveguides) that is relatively established, and they demonstrate a system comprising 11 tunable waveguides, which is not sufficient to access new regimes or stimulate interesting new phenomena.*

Response: As we mentioned in our reply to the comment above, our work is a key step towards versatile and scalable photonic integrated circuits based on continuously coupled waveguides. We believe this makes it a significant advancement to the existing knowledge in this area. Specifically, this platform provides substantial progress toward new applications including:

- implementation of universal unitaries with improved fabrication error tolerance and significantly reduced bending loss [7–11]
- high-speed multimode routing and mode division multiplexing [12–15],
- Hamiltonian continuous-time evolutions implementation with precisely controlled Hamiltonian parameters, as an alternative approach to the discrete implementation [16],
- simulations of higher-dimensional systems such as in recent proposal [17, 18]
- possibility of demonstrating more complex experiments [17–20], as well as continuous simulation of time for arbitrary Hamiltonians time evolution dynamics.

Comment: *The key problem is understanding how far we can go in this direction. In this respect, this second version of the manuscript does not provide any new element. In my previous report, I asked to provide more details, for example, a reasonable number of waveguides that one could implement, but this question remained unanswered.*

Response: We apologize for not addressing this point in our previous response.

In theory, 1000's waveguides at a coupling distance of 10 μm can be patterned on a 6" LN wafer. Even with the electrical wiring and fan-in fan-out of the waveguide to match fiber arrays, it is feasible to produce a device with 100's of waveguides, which is on par with the number of modes implemented in large-scale photonic circuits reported to date, e.g. in [21].

We have added the following sentence in the 'Discussion' section:

In theory, thousands of waveguides at a coupling distance of 10 μm can be patterned on a 6-inch lithium niobate wafer. Even with the electrical wiring and fan-in fan-out of the waveguide to match fiber arrays, it is feasible to produce a device with hundreds of waveguides.

Comment: *I find it somewhat curious that in the Discussion section, the authors suggest that the scalability of their approach may be achieved in the thin-film LiNbO platform, where a more compact design could be implemented. This is strange, not only because this is not the approach investigated here, but because on page 2 the same authors write: "To avoid voltage breakdown through air and glass, the electrodes must be patterned at sufficient separation. This is possible because the large cross-section of the mode in a proton-exchange waveguide leads to substantial coupling between waveguides more than 10 μm apart," which seems to indicate an advantage of well-separated waveguides. Furthermore, implementing this idea in a different platform would certainly require addressing new experimental issues, (e.g. electrode configuration), which are not mentioned at all.*

Response: We thank the referee for pointing this out.

The proton-exchanged lithium niobate (PE:LN) is an established electro-optic technology that is used here to demonstrate the concept. However, we believe that thin film lithium niobate (LNOI) can also be used to implement this device. In particular, z-cut LNOI, where the electric field is directed vertically, requires a smaller footprint and significantly lower driving voltage [22]. This could enable our device to be implemented in LNOI, but this requires further investigation.

We have improved the following sentence in the in the 'Discussion' section to clarify this point:

In particular, the z-cut thin film lithium niobate platform may allow reducing the footprint—enabling the fabrication of cascaded RWAs—while keeping the advantage of electro-optic performance including spacing the electrodes, high-speed operations and low driving voltage [22, 23].

Comment: *Another concern raised in my previous report was related to universality. In their reply, the authors state that "our platform, when cascaded in series, could implement arbitrary unitaries [6]" While this statement holds true in theory, the practical implications remain unclear. How many distinct sections are necessary? What length must each section have? Is the approach feasible in terms of minimizing losses? Unfortunately, the authors' response does not offer sufficient information to establish the scalability and universality of their approach. Thus, the impact of their experimental result, which is one of the main issues, remains unclear.*

Response: The main goal of our work is to demonstrate the first prototype of a large-scale reconfigurable waveguide array (RWA), a technology that so far an only been based on static devices. The discussion on scalability and universality has been done in [8], which:

- shows that the number of sections required increases linearly with the number of modes,

- motivates the waveguide array architecture by discussing that one of the main sources of loss of photonics circuits comes from bends, also one of the motivations of Clements et al. work [24, 25], which improved on work by Reck et al. [26]. The number of in-and-out bending sections one photon can experience in Clements' scheme is $4N$ (where N is the number of waveguides), while the number is $2N$ in waveguide array-based scheme [8].
- proves that such an architecture is universal.

At this stage, there's no design guideline about the length required for the coupled region for implementing universal operations, as it's a topic of ongoing research in the field of controllability.

While universality is not the main claim of our manuscript, we agree with the referee that the feasibility of implementing arbitrary unitaries with our device requires further work, which is out of the scope of this project.

We have added the following sentence in the in the 'Discussion' section:

Compared with the scheme by Clements *et al.* [24], the waveguide array-based scheme [8] incurs lower bending losses due to the half number of bending sections the photons will experience.

Comment: *Previously, I highlighted that the experimental results presented in this work are purely classical in nature. I appreciate that the authors made this clear in this second version of the manuscript by removing/modifying some of their statements. However, their assertion, "Furthermore, a classical (laser light) characterization of the Hamiltonian completely predicts the device performance with quantum states," can be misleading. While it is indeed possible to determine the device's transfer function using classical light, it is important to recognize that the device performance is influenced also by the quantum state and the specific experimental conditions. For instance, the spectral properties of the photons will play a crucial role in determining the visibility of quantum interference, depending upon the waveguide dispersion.*

Response: Characterizing linear photonic devices with laser light at a wavelength close to that of the single photons used in a quantum experiment is the standard practice [1, 27, 28] and gives full information about the quantum process of the devices as shown in [29, 30]. A small spectral difference between laser light and single photon sources will give a negligible change in the device Hamiltonian.

We have improved the following sentence in the 'Discussion' section:

Furthermore, laser light characterization of the Hamiltonian completely predicts the quantum process of linear photonic devices [29, 30], and the small spectral difference between laser light and single photon sources will give a negligible change in the device Hamiltonian.

Comment: *It should be noted that the experiment described in the paper is conducted at a single wavelength (808 nm), a detail which regrettably goes unmentioned in the main text.*

Response: We thank the referee for pointing this out. We have added the following text at the end of the 'Waveguide array overview and modeling' section:

The light source used in this work is a fiber-coupled polarized 808 nm laser diode.

Comment: *Furthermore, the fidelities measured in this work may not be meaningful in a quantum experiment. Indeed, the device is burdened with significant insertion losses, which could render quantum experiments impractical. The prospect of low detection rates, for example, could necessitate excessively lengthy experiments, potentially impacting the device's stability in ways not thoroughly investigated in this study. Therefore, if the intention is to assert the device's viability for quantum experiments, it becomes imperative to conduct such experiments. This is because a host of complexities exist beyond a mere linear characterization using continuous-wave laser light.*

Response: We disagree with the reviewer. As discussed in reply to the comment above, laser light characterization gives us full knowledge of the quantum process of linear photonic devices. This fact makes this comment by the reviewer not correct.

The total loss for this device, including insertion and transmission loss was 90%. It is possible to perform quantum experiments with such loss, as it was demonstrated in the first two-photon quantum walk in a static waveguide array [1], and many others. To enhance the total transmission of the device, for example for experiments using more than two photons, the insertion loss can be reduced by various technologies, ranging from on-chip components to engineered fibers [2–6].

- a periodically segmented waveguide coupler in PE:LN [2]
- polishing optical facets [3]
- soft-proton-exchange taper that supports lossless mode-shape transformation [4]
- lensed fiber [5]
- mode-selective coupler in LiNbO₃:Ti optical waveguides with coupling efficiencies larger than 90% [6]

We have improved the following text and added the detail of full insertion loss in the ‘Measurement and control setup’ section in the Supplementary Methods:

A polarized 808 nm laser and multi-channel fiber-coupled high-speed optical power meter were used for the output measurements. Polarization-maintaining fiber (PMF) arrays with 127 μm pitch were used for butt-coupling to the chip at the input and output sides with a coupling loss of 4.9 dB per facet (68%), caused by fiber-waveguide mode mismatch. A total power loss, including the fiber-to-chip coupling and propagation loss, of 9.8 dB (90%) was measured. This is sufficient for two-photon quantum experiments, as demonstrated in the first two-photon quantum walk in a static waveguide array [1]. To enhance the total transmission of the device, for example for experiments using more than two photons, the insertion loss can be reduced by various technologies, ranging from on-chip components to engineered fibers, such as high-index fibers that reduce the mode field diameter of the fiber [2–6].

-
- [1] Alberto Peruzzo, Mirko Lobino, Jonathan C. F. Matthews, Nobuyuki Matsuda, Alberto Politi, Konstantinos Poullos, Xiao-Qi Zhou, Yoav Lahini, Nur Ismail, Kerstin Wörhoff, Yaron Bromberg, Yaron Silberberg, Mark G. Thompson, and Jeremy L. O'Brien. Quantum walks of correlated photons. *Science*, 329(5998):1500–1503, 2010. Publisher: American Association for the Advancement of Science.
- [2] Pierre Aschieri and Marc P de Micheli. Highly efficient coupling in lithium niobate photonic wires by the use of a segmented waveguide coupler. *Applied optics*, 50(21):3885–3888, Jul 2011.
- [3] E.L. Wooten, K.M. Kissa, A. Yi-Yan, E.J. Murphy, D.A. Lafaw, P.F. Hallemeier, D. Maack, D.V. Attanasio, D.J. Fritz, G.J. McBrien, and D.E. Bossi. A review of lithium niobate modulators for fiber-optic communications systems. *IEEE Journal of Selected Topics in Quantum Electronics*, 6(1):69–82, January 2000. Conference Name: IEEE Journal of Selected Topics in Quantum Electronics.
- [4] Davide Castaldini, Paolo Bassi, Sorin Tascu, Pierre Aschieri, Marc P. De Micheli, and Pascal Baldi. Soft-proton-exchange tapers for low insertion-loss LiNbO₃ devices. *Journal of Lightwave Technology*, 25(6):1588–1593, 2007.
- [5] Optical waveguides and devices in lithium niobate by the proton exchange process. - core. <https://core.ac.uk/works/27528713>, 2023.
- [6] Daniel Runde, Stefan Breuer, and Detlef Kip. Mode-selective coupler for wavelength multiplexing using linbo3:ti optical waveguides. *Open Physics*, 6(3):588–592, 2008.
- [7] J. Petrovic and J. J. P. Veerman. A new method for multi-bit and qudit transfer based on commensurate waveguide arrays. *Annals of Physics*, 392:128–141, May 2018.
- [8] M. Yu. Saygin, I. V. Kondratyev, I. V. Dyakonov, S. A. Mironov, S. S. Straupe, and S. P. Kulik. Robust architecture for programmable universal unitaries. *Phys. Rev. Lett.*, 124:010501, Jan 2020.
- [9] Jovana Petrovic, Jelena Krsic, Peter J. J. Veerman, and Aleksandra Maluckov. A new concept for design of photonic integrated circuits with the ultimate density and low loss, 2021.
- [10] N. N. Skryabin, N. N. Skryabin, I. V. Dyakonov, M. Yu Saygin, and S. P. Kulik. Waveguide-lattice-based architecture for multichannel optical transformations. *Optics Express*, 29(16):26058–26067, 2021.
- [11] Ryota Tanomura, Rui Tang, Toshikazu Umezaki, Go Soma, Takuo Tanemura, and Yoshiaki Nakano. Scalable and Robust Photonic Integrated Unitary Converter Based on Multiplane Light Conversion. *Physical Review Applied*, 17(2):024071, February 2022.
- [12] Alastair Kay. Perfect state transfer: Beyond nearest-neighbor couplings. *Phys. Rev. A*, 73:032306, Mar 2006.
- [13] Enrico Compagno, Leonardo Banchi, and Sougato Bose. Toolbox for linear optics in a one-dimensional lattice via minimal control. *Physical Review A*, 92(2):022701, 2015.
- [14] Philippe Velha, Isabella Cerutti, and Nicola Andriolli. Crosstalk and ber performance of closely-spaced silicon-on-insulator waveguide arrays. *Optics Communications*, 437:214–218, 2019.
- [15] N. N. Skryabin, I. V. Dyakonov, M. Yu. Saygin, and S. P. Kulik. Waveguide-lattice-based architecture for multichannel optical transformations. *Opt. Express*, 29(16):26058–26067, Aug 2021.
- [16] Nicholas C. Harris, Gregory R. Steinbrecher, Mihika Prabhu, Yoav Lahini, Jacob Mower, Darius Bunandar, Changchen Chen, Franco N. C. Wong, Tom Baehr-Jones, Michael Hochberg, Seth Lloyd, and Dirk Englund. Quantum transport simulations in a programmable nanophotonic processor. *Nature Photonics*, 11(7):447–452, 2017.
- [17] Stefano Longhi. Photonic simulation of giant atom decay. *Optics Letters*, 45(11):3017–3020, June 2020.
- [18] Lukas J. Maczewsky, Kai Wang, Alexander A. Dovgiy, Andrey E. Miroshnichenko, Alexander Moroz, Max Ehrhardt, Matthias Heinrich, Demetrios N. Christodoulides, Alexander Szameit, and Andrey A. Sukhorukov. Synthesizing multi-dimensional excitation dynamics and localization transition in one-dimensional lattices. *Nature Photonics*, 14(2):76–81, February 2020.
- [19] Miguel A. Bandres, Oded Zilberberg, and Andrey Sukhorukov. Special topic on synthetic gauge field photonics. *APL Photonics*, 7(5):050401, 05 2022.
- [20] Javier del Pino and Oded Zilberberg. Dynamical gauge fields with bosonic codes. *Phys. Rev. Lett.*, 130:171901, Apr 2023.
- [21] Y. Lahini, R. Pugatch, F. Pozzi, M. Sorel, R. Morandotti, N. Davidson, and Y. Silberberg. Observation of a localization transition in quasiperiodic photonic lattices. *Physical Review Letters*, 103(1):013901, 2009.
- [22] Mian Zhang, Cheng Wang, Prashanta Kharel, Di Zhu, and Marko Lončar. Integrated lithium niobate electro-optic modulators: when performance meets scalability. *Optica*, 8(5):652–667, May 2021.
- [23] Inna Krasnokutskaya, Jean-Luc J. Tambasco, and Alberto Peruzzo. Tunable large free spectral range microring resonators in lithium niobate on insulator. *Scientific Reports*, 9(1):11086, 06 2018.
- [24] William R. Clements, Peter C. Humphreys, Benjamin J. Metcalf, W. Steven Kolthammer, and Ian A. Walmsley. Optimal design for universal multiport interferometers. *Optica*, 3(12):1460–1465, 2016.
- [25] Roel Burgwal, William R. Clements, Devin H. Smith, James C. Gates, W. Steven Kolthammer, Jelmer J. Renema, and Ian A. Walmsley. Using an imperfect photonic network to implement random unitaries. *Opt. Express*, 25(23):28236–28245, Nov 2017.
- [26] Michael Reck, Anton Zeilinger, Herbert J. Bernstein, and Philip Bertani. Experimental realization of any

- discrete unitary operator. *Physical Review Letters*, 73(1):58–61, 1994.
- [27] Hsuan-Hao Lu, Joseph M. Lukens, Nicholas A. Peters, Ogaga D. Odele, Daniel E. Leaird, Andrew M. Weiner, and Pavel Lougovski. Electro-optic frequency beam splitters and tritters for high-fidelity photonic quantum information processing. *Phys. Rev. Lett.*, 120:030502, Jan 2018.
- [28] Matthew A. Broome, Alessandro Fedrizzi, Saleh Rahimi-Keshari, Justin Dove, Scott Aaronson, Timothy C. Ralph, and Andrew G. White. Photonic boson sampling in a tunable circuit. *Science*, 339(6121):794–798, 2013.
- [29] Mirko Lobino, Dmitry Korystov, Connor Kupchak, Eden Figueroa, Barry C. Sanders, and A. I. Lvovsky. Complete characterization of quantum-optical processes. *Science*, 322(5901):563–566, 2008.
- [30] Saleh Rahimi-Keshari, Matthew A Broome, Robert Fickler, Alessandro Fedrizzi, Timothy C Ralph, and Andrew G White. Direct characterization of linear-optical networks. *Optics express*, 21(11):13450–13458, 2013.

REVIEWER COMMENTS

Reviewer #3 (Remarks to the Author):

I am writing to submit my third-round review of the manuscript titled "Programmable High-Dimensional Hamiltonian in a Photonic Waveguide Array."

In this third version of the manuscript, the authors have not addressed all of my comments. The most critical concern revolves around the significance of their work. Manuscripts published in *Nature Communications* should showcase substantial advancements in one or more fields, capable of inspiring further valuable research. This journal holds a significant influence in shaping the future research directions across various domains. Therefore, it is imperative that the authors present compelling arguments and robust theoretical/experimental evidence to support their claims. Regrettably, after the third resubmission, the authors have not effectively demonstrated the relevance of their findings in the realm of reconfigurable photonic architectures for photonic computing, whether in classical or quantum contexts. This skepticism is also shared by Reviewer #2, and I find it hard to argue otherwise.

Firstly, the authors have not tackled the issue of scalability. They simply assert the potential implementation of a greater number of waveguides using another platform, TFLN, without presenting any experimental results.

Secondly, the authors have provided a classical characterization of their system to infer its quantum performance but have failed to offer any quantum results to substantiate their claims. Their response indicates that the experiment is feasible with their chip, so why have they not conducted it? Furthermore, in the latest version of the manuscript, the authors state, "A total power loss, including the fiber-to-chip coupling and propagation loss, of 9.8 dB (90%) was measured. This is sufficient for two-photon quantum experiments, as demonstrated in the first two-photon quantum walk in a static waveguide array [1]." This statement is somewhat underwhelming. It is reasonable to expect that after more than twelve years, there should be more than just the promise of a two-photon experiment.

In their manuscript, the authors suggest a potential future direction for reconfigurable photonics, but they have not substantiated it with evidence. Consequently, I cannot endorse the publication of this work in *Nature Communications*.

Referee 3

Comment 1 - Reviewer #3 says: "The most critical concern revolves around the significance of their work" and "This skepticism is also shared by Reviewer #2, and I find it hard to argue otherwise".

Response 1 - We understand the concern of Reviewer #3 **but his concern is not shared by Reviewer #2** who wrote the following remarks after the first resubmission: "the Authors' reply together with the revised manuscript make me appreciate more clearly the substantial technological advancements that the Authors have achieved", "As a matter of fact, the reconfiguration of a waveguide array had never been demonstrated with this degree of detail and within the coupling region", "these results are indeed significant and impactful in the wider field of photonics", and "I can now support the publication of this manuscript in Nature Communications". These are the same reasons why we believe our paper is relevant.

Comment 2 - Reviewer #3 says: "the authors have not tackled the issue of scalability. They simply assert the potential implementation of a greater number of waveguides using another platform, TFLN, without presenting any experimental results."

Response 2 – We addressed the scalability of our platform based on proton exchanged waveguides in the discussion section with an estimation of the maximum number of waveguides that can be patterned on a wafer. Regarding the thin-film-lithium-niobate waveguides, references from [59 to 64] offer a range of experimental results with this platform showing its potential and scalability. It is not the aim of this paper to present results with TFLN devices. Here we show the reconfigurability potential of electro-optical devices instead of thermo-optical ones (see comments from Reviewer #2 in Response 1).

Comment 3 - Reviewer #3 says: "the authors have provided a classical characterization of their system to infer its quantum performance but have failed to offer any quantum results to substantiate their claims. Their response indicates that the experiment is feasible with their chip, so why have they not conducted it?"

Response 3 – As stated in the discussion section of our revised paper and references therein, device characterization with laser light offers full knowledge of the quantum process performed by the devices. At present, we are in the process of moving our labs and we are not equipped for performing measurements with single photons. However, as discussed above and our significant experience, we don't expect any issues with performing a quantum experiment.

Comment 4 - Furthermore, in the latest version of the manuscript, the authors state, "A total power loss, including the fiber-to-chip coupling and propagation loss, of 9.8 dB (90%) was measured. This is sufficient for two-photon quantum experiments, as demonstrated in the first two-photon quantum walk in a static waveguide array [1]." This statement is somewhat underwhelming. It is reasonable to expect that after more than twelve years, there should be more than just the promise of a two-photon experiment.

Response 4 – Losses in the device can be mitigated with optimization of the fabrication process or working at telecom wavelength. We addressed this comment adding the following sentence to the "Measurement and control setup" section of the Supplementary Materials:

"Losses can be mitigated by improving the fabrication process and/or working at telecom wavelength where propagation and coupling losses are lower. [2]"

Comment 5 – "In their manuscript, the authors suggest a potential future direction for reconfigurable photonics, but they have not substantiated it with evidence."

Response 5 – We have already addressed this comment in the discussion section with references 41-43 and 20, 22, 23, 44, 45.

REVIEWERS' COMMENTS

Reviewer #3 (Remarks to the Author):

I am writing to submit my fourth-round review of the manuscript titled "Programmable High-Dimensional Hamiltonian in a Photonic Waveguide Array."

Since my very first report, I pointed out that these experimental results are solid and convincing to demonstrate that their platform is flexible and reconfigurable in a predictive way. However, I expressed serious doubts that the manuscript contains enough results to support the authors' claims about the real potential of this platform in quantum photonics. These doubts persist, as the authors have presented only experimental data related to classical device characterization, with no quantum results, and the device's complexity remains comparatively modest in comparison to other platforms showcasing quantum experiments.

Throughout my previous reviews, I raised various questions regarding scalability, complexity, and losses. The authors' responses consistently pointed to potential solutions that lacked empirical validation, relying on promises rather than demonstrated outcomes. For instance:

-Scalability: "There is no fundamental problem with building RWA with hundreds of spatial modes and cascading them to implement larger scale circuits" Yet, this has not been demonstrated.

-Complexity: "Our platform, when cascaded in series, could implement arbitrary unitaries." However, this has not been demonstrated in practice.

-Losses impact: "Losses can be mitigated by improving the fabrication process and/or working at telecom wavelength where propagation and coupling losses are lower." This has not been demonstrated in this kind of implementation.

I understand that addressing these concerns may involve substantial research efforts beyond the manuscript's scope. Nevertheless, rather than dismissing these issues, it would be more beneficial to alert readers to potential challenges in further developing this platform.

More importantly, as I mentioned in my previous report, I strongly believe that a quantum experiment is essential to substantiate the authors' claims and the significance of their findings. In their response, the authors mention, "We are in the process of moving our labs and we are not equipped for performing measurements with single photons. However, as discussed above and our significant experience, we don't expect any issues with performing a quantum experiment." Regrettably, this response cannot be deemed acceptable. It would be surprising to encounter a paper in journals like *Nature Communications*, or any reputable journal, where authors refrain from conducting a pivotal experiment due to laboratory issues, yet have their manuscript accepted and published based on their reputation alone.

Should the authors provide a quantum characterization of their device, I would be more than willing to re-evaluate the manuscript.

Referee 3

Comment 1: on “Scalability: “There is no fundamental problem with building RWA with hundreds of spatial modes and cascading them to implement larger scale circuits” Yet, this has not been demonstrated.”

Comment 2: on “Complexity: “Our platform, when cascaded in series, could implement arbitrary unitaries.” However, this has not been demonstrated in practice.”

Response: To address comments 1 and 2, we added the following reference to the first sentence of the discussion section, where cascaded devices have been demonstrated on a waveguide geometry very close to the proton exchange one.

Reconfigurable waveguide arrays are a powerful tool for optical signal routing, implementing time-evolution Hamiltonians to simulate quantum materials and phenomena, and can be cascaded [57] to realize arbitrary unitaries.

[57] Simone Atzeni, Adil S. Rab, Giacomo Corrielli, Emanuele Polino, Mauro Valeri, Paolo Mataloni, Nicolò Spagnolo, Andrea Crespi, Fabio Sciarrino, and Roberto Osellame, "Integrated sources of entangled photons at the telecom wavelength in femtosecond-laser-written circuits," *Optica* 5, 311-314 (2018)

Comment 3: Review #3 says: “Losses impact: “Losses can be mitigated by improving the fabrication process and/or working at telecom wavelength where propagation and coupling losses are lower.” This has not been demonstrated in this kind of implementation.”

Response: We previously addressed this comment showing that improving losses is possible and have been experimentally demonstrated. We added one more reference to the following sentence:

Additionally, losses can be mitigated by improving the fabrication process and/or working at telecom wavelength where propagation and coupling losses are lower [2, 46]